# EGF-dependent re-routing of vesicular recycling switches spontaneous phosphorylation suppression to EGFR signaling

Martin Baumdick[1†], Yannick Brüggemann[1,2†], Malte Schmick[1†], Georgia Xouri[1†], Ola Sabet[1], Lloyd Davis[3], Jason W Chin[3], Philippe IH Bastiaens[1,2*]

[1]Department of Systemic Cell Biology, Max Planck Institute of Molecular Physiology, Dortmund, Germany; [2]Faculty of Chemistry and Chemical Biology, Technical University of Dortmund, Dortmund, Germany; [3]Medical Research Council Laboratory of Molecular Biology, Cambridge, United Kingdom

**Abstract** Autocatalytic activation of epidermal growth factor receptor (EGFR) coupled to dephosphorylating activity of protein tyrosine phosphatases (PTPs) ensures robust yet diverse responses to extracellular stimuli. The inevitable tradeoff of this plasticity is spontaneous receptor activation and spurious signaling. We show that a ligand-mediated switch in EGFR trafficking enables suppression of spontaneous activation while maintaining EGFR's capacity to transduce extracellular signals. Autocatalytic phosphorylation of tyrosine 845 on unliganded EGFR monomers is suppressed by vesicular recycling through perinuclear areas with high PTP1B activity. Ligand-binding results in phosphorylation of the c-Cbl docking tyrosine and ubiquitination of the receptor. This secondary signal relies on EGF-induced EGFR self-association and switches suppressive recycling to directional trafficking. The re-routing regulates EGFR signaling response by the transit-time to late endosomes where it is switched-off by high PTP1B activity. This ubiquitin-mediated switch in EGFR trafficking is a uniquely suited solution to suppress spontaneous activation while maintaining responsiveness to EGF.

*For correspondence: philippe. bastiaens@mpi-dortmund.mpg.de

†These authors contributed equally to this work

**Competing interests:** The authors declare that no competing interests exist.

## Introduction

Signaling by the epidermal growth factor receptor (EGFR) converts diverse external stimuli into specific cellular responses. EGFR signaling is implicated in embryonic development, tissue homeostasis and wound healing (*Yu et al., 2010*; *Sibilia et al., 2007*), while EGFR overexpression and hyper-activation through genetic alterations have been linked to malignant transformation (*Rowinsky, 2004*). Mutation-induced loss of autoinhibitory interactions or enhanced receptor expression levels, frequently elevate the basal phosphorylation and activation status of EGFR (*Arteaga and Engelman, 2014*). The autoinhibitory interactions include the tethered conformation of the extracellular domain (*Ferguson et al., 2003*) as well as receptor-membrane interactions and the local intrinsic disorder of the αC helix in the N-lobe of the kinase domain (*Arkhipov et al., 2013*; *Endres et al., 2013*; *Shan et al., 2012*). This creates an energy barrier for EGFR self-association that is surpassed upon ligand stimulation. Ligand binding leads to receptor dimerization (*Yarden and Schlessinger, 1987*; *Cochet et al., 1988*) and the formation of an asymmetric dimer of the intracellular kinase domains (*Zhang et al., 2006*). This triggers phosphorylation *in trans* of regulatory and signaling tyrosine residues in the intracellular part of the receptor, and a subsequent recruitment of adaptor proteins that contain Src homology 2 domains (SH2) or phosphotyrosine-binding domains (PTB) such as c-Cbl

**eLife digest** In living tissue, the ability of individual cells to grow is influenced by signal molecules in the environment around each cell. For example, after an injury, a molecule called epidermal growth factor can stimulate cells to grow to repair the wound. Epidermal growth factor binds to and activates a receptor protein called EGFR, which faces outwards from the cell surface. However, this signal needs to be switched off again afterwards to prevent the cells from growing too much.

Epidermal growth factor activates EGFR by triggering a process called "autophosphorylation", in which EGFR attaches molecules called phosphates to itself. To quench the signal, EGFRs that are bound to growth factors are removed from the cell surface and taken into the cell in small membrane bubbles called vesicles. Enzymes called phosphatases near the cell nucleus remove the phosphate groups and thereby switch the receptors off, before the receptors are ultimately destroyed. However, EGFR autophosphorylation can also happen spontaneously in the absence of growth factor, so it was not clear how the cell is able to distinguish between this spontaneous activation and a genuine signal.

Baumdick, Brüggemann, Schmick, Xouri et al. used biochemical techniques to address this question. The experiments show that EGFRs that have become spontaneously active are also removed from the cell surface in vesicles. However, unlike the EGFRs that are bound to growth factors, the spontaneously active receptors are recycled back to the membrane. On the way, their activity is also switched off by encountering phosphatases so that they are not active when they reach the cell surface again.

The experiments also show that EGFRs are targeted for destruction by the presence of a tag called ubiquitin, which is added to the receptor in response to the binding of growth factor. Therefore, Baumdick et al.'s findings show that epidermal growth factor controls a switch that alters the way active EGFRs are processed in cells. This system acts to suppress the spontaneous activation of EGFRs, whilst maintaining the ability of the cell to respond to epidermal growth factor. The next challenge is to understand how the location of the phosphatases inside the cell influences when and how the EGFRs respond to this external signal.

(Y1045) or Grb2 (Y1068 and Y1086) (*Ushiro and Cohen, 1980*; *Moran et al., 1990*; *Levkowitz et al., 1998*; *Waterman et al., 2002*; *Lemmon and Schlessinger, 2010*).

Despite these EGFR structure intrinsic safeguards, the receptor can still attain an active conformation in the absence of ligand due to thermal fluctuations (*Lemmon and Schlessinger, 2010*), necessitating only low protein tyrosine phosphatase (PTP) activity to suppress phosphorylation due to this 'leaky' kinase activity. However, phosphorylation of the conserved regulatory tyrosine Y845 in the activation loop of the EGFR kinase domain leads to an acceleration of its phosphorylation, potentiating EGFR kinase activity in an autocatalytic fashion (*Shan et al., 2012*). Such an autocatalytic activation system that is coupled to PTP activity, by for example a double negative feedback, offers robustness against biological noise and conveys external stimuli into threshold-activated responses (*Grecco et al., 2011*). Autocatalysis can lead to amplified self-activation of the receptor in the absence of a cognate ligand (*Verveer, 2000*; *Endres et al., 2013*), requiring high PTP activity at the plasma membrane (PM) to suppress. Such PTPs that act on EGFR with high catalytic efficiency (~2 orders of magnitude higher than EGFR) are PTP1B and TCPTP (*Zhang et al., 1993*; *Romsicki et al., 2003*; *Fan et al., 2004*). These PTPs are, however, segregated from the PM by association with the cytoplasmic membrane leaflet of the endoplasmic reticulum (ER), and therefore mostly dephosphorylate endocytosed ligand-bound EGFR.

After ligand binding, endocytosed receptor-ligand complexes contained in clathrin-coated vesicles (CCVs) enter early endosomes (EEs) by fusion (*Vieira et al., 1996*; *Bucci et al., 1992*; *Goh and Sorkin, 2013*), further maturing in the perinuclear area to late endosomes (LEs) and eventually fusing to lysosomes where receptors are degraded (*Rink et al., 2005*; *Ceresa, 2006*; *Vanlandingham and Ceresa, 2009*; *Levkowitz et al., 1999*). Although EGFR vesicular trafficking was extensively studied after ligand stimulation, little is known about the role of vesicular trafficking in suppressing spontaneous EGFR activation as well as regulating its signaling response. To assess

how vesicular membrane dynamics modulates spontaneous and ligand-induced phosphorylation of EGFR, we studied three phosphorylation sites on EGFR with distinct functionality: 1) Y845—a regulatory autocatalytic tyrosine whose phosphorylation increases EGFR activity (*Shan et al., 2012*), 2) Y1045—a site that upon phosphorylation affects vesicular trafficking of EGFR by binding the E3 ligase c-Cbl that ubiquitinates the receptor (*Levkowitz et al., 1998*), and 3) Y1068—a site that upon phosphorylation binds the adapter Grb2 via its SH2 domain to propagate signals in the cell (*Okutani et al., 1994*).

We show that spontaneously and ligand-induced EGFR activation gives rise to distinct molecular states that are recognized and processed differently by the endocytic machinery. While unliganded monomeric receptors continuously recycle to the PM to suppress autocatalytic activation, ligand-bound dimeric receptors are ubiquitinated by the E3-ligase c-Cbl that commits them to unidirectional vesicular trafficking toward lysosomes. This route through perinuclear endosomes enables their efficient dephosphorylation by high local PTP activity to produce a finite signaling response to growth factors. We demonstrate by a compartmental model that ligand-responsive EGFR signaling can only occur in conjunction with suppression of spontaneous autocatalytic EGFR activation if a ligand-induced switch in EGFR trafficking changes its cyclic interaction with spatially partitioned PTPs to a sustained one.

## Results

### The dependence of spontaneous EGFR phosphorylation on its expression level

To investigate how EGFR auto-phosphorylation depends on its cell surface density, we quantified the relative phosphorylation (pY/EGFR) of three tyrosine residues with distinct regulatory functionality of autocatalysis, signaling, and trafficking in single COS-7 cells as a function of EGFR-mCitrine expression level. The variance in ectopic expression of EGFR-mCitrine was thereby exploited to sample a broad range of receptor expression levels. The EGFR-mCitrine expression level in single cells was determined relative to endogenous EGFR by an independent immunofluorescence experiment where the level of endogenously expressed EGFR was quantified from the abscissa-intercept of a linear fit to an EGFR-mCitrine intensity versus anti-EGFR antibody intensity plot (*Figure 1—figure supplement 1A,B*). This analysis showed that most COS-7 cells expressed EGFR-mCitrine at similar level as endogenous EGFR, whereas the expression varied by a factor 6 (from ~0.5–3) (*Figure 1A*, X-axis). The phosphorylation level of the autocatalytic Y845 site as well as the Cbl (Y1045) and Grb2 (Y1068) docking sites was determined for single cells as well as for cell populations by quantifying immunofluorescence staining (*Figure 1A,B*, *Figure 1—figure supplement 1C*) and Western blots (*Figure 1C–E*, *Figure 1—figure supplement 2, 3*), respectively. To determine the levels of spontaneous EGFR phosphorylation by Western blot analysis, its expression was controlled by the amount of transfected EGFR-mCitrine encoding cDNA and quantified by the EGFR-mCitrine band intensity relative to the band of the maximally applied cDNA amount (3 µg) (*Figure 1C–E*, *Figure 1—figure supplement 2, 3*). Single-cell immunofluorescence analyses showed that a wide range of auto-phosphorylation levels occur for similar EGFR expression in individual cells (*Figure 1A*). Despite this cell-to-cell variance in response, clear trends in the average phosphorylation of the individual sites as function of EGFR-mCitrine expression were observed. The average amount of spontaneously phosphorylated Y845 on EGFR increased ~ sixfold with the density of EGFR in cells (*Figure 1A*, left panel). Upon EGF-stimulation, the relative phosphorylation of Y845 increased, exhibiting a clear dependency on the EGFR expression levels as well (elevated by ~ threefold for a ~ sixfold increase in EGFR), corroborating the autocatalytic function of Y845 (*Shan et al., 2012*). The c-Cbl docking site Y1045 exhibited a very different response profile: spontaneous phosphorylation was weakly increased with EGFR expression (~ twofold increase), whereas EGF stimulation clearly resulted in phosphorylation levels that were independent of EGFR expression (*Figure 1A*, middle panel). On the other hand, the average amount of spontaneously phosphorylated Y1068 strongly increased (by ~ eightfold) with the EGFR density in cells (*Figure 1A*, right panel). However, in contrast to Y845, EGF stimulation increased its average phosphorylation independent of EGFR expression (*Endres et al., 2013*).

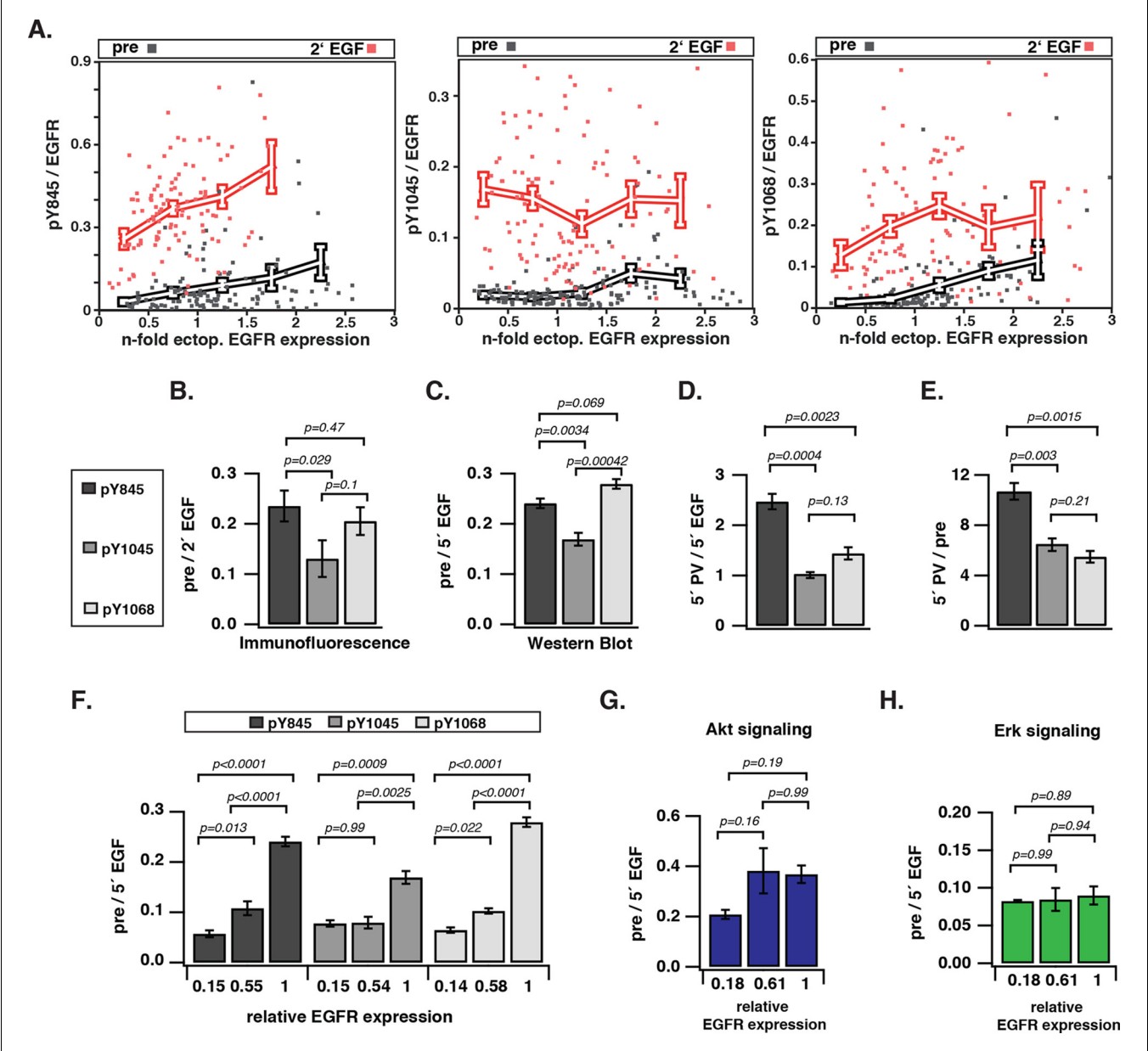

**Figure 1.** Spontaneous tyrosine phosphorylation in response to EGFR expression. (**A**) Scatter plots show the relative phosphorylation (mean fluorescence intensity of pYN-antibody/EGFR-mCitrine) of three tyrosine residues (N= 845, 1045, or 1068) versus EGFR-mCitrine expression in multiples of mean endogenous EGFR levels (see *Figure 1—figure supplement 1A,B*). Points (black: pre-; red: 2-min post-stimulation with 100 ng/ml EGF) represent single COS-7 cells (see *Figure 1—figure supplement 1C*) and thick lines indicate mean values of binned data. (**B,C**) Spontaneous phosphorylation of Y845, Y1045 and Y1068. (**B**) Immunofluorescence data showing mean fluorescence intensity of pYN-antibody over EGFR-mCitrine (pre) normalized to mean fluorescence intensity of pYN-antibody over EGFR-mCitrine, 2-min post-stimulation with 100 ng/ml EGF (2' EGF). YN(number of cells pre, 2' EGF): Y845(119, 108); Y1045(145, 109); Y1068(127, 87). (**C**) Western blot analysis of COS-7 lysates probed for anti-GFP antibody and either anti-pY845 (n=5 blots), anti-pY1045 (n=3) or anti-pY1068 (n=3) show pYN-antibody band over anti-GFP band (pre) normalized to the same fraction 5-min post-stimulation with 100 ng/ml EGF (5' EGF). Data correspond to the highest levels of expressed EGFR-mCitrine (3 μg cDNA, *Figure 1—figure supplement 2*). (**D, E**) Autonomous phosphorylation of Y845, Y1045, and Y1068 upon pervanadate (PV) treatment. Western blot analysis of COS-7 lysates probed for anti-pY845, anti-pY1045, anti-pY1068 and anti-GFP antibody (all n=3 blots). (**D**) pYN-antibody band over anti-GFP band 5 min post-addition of PV (0.33 mM) normalized to relative phosphorylation 5-min post-stimulation with 100 ng/ml EGF (*Figure 1—figure supplement 3*). (**E**) pYN-antibody band over anti-GFP band 5-min post-addition of PV (0.33 mM) normalized to relative phosphorylation before stimulation (pre). (**F**) Dependence of EGFR phosphorylation on its expression level. Western blot analysis of COS-7 lysates transfected with increasing amounts (0.5, 1.5, and 3 μg cDNA) of EGFR-mCitrine. Blots were probed for anti-pY845 (n=5 blots), anti-pY1045 (n=3), anti-pY1068 (n=3), and anti-GFP for EGFR-mCitrine (*Figure 1—figure supplement 2*). The ordinate displays the fraction of EGFR-band over tubulin band in each lane relative to 3 μg cDNA and bar

*Figure 1. continued on next page*

*Figure 1. Continued*

diagram shows relative EGFR phosphorylation of the three tyrosine residues (see 'Materials and methods'). (G, H) Dependence of Akt and Erk activation on EGFR expression level. Western blots of COS-7 lysates transfected with increasing amounts (0.5, 1.5, and 3 µg cDNA) of EGFR-mCitrine and probed for phosphorylated Ser473 on Akt, total Akt, phosphorylated Thr202/Tyr204 on Erk1/2, and total Erk1/2 levels. Data represents 'ratio of fractions' of either Akt phosphorylation (E) or Erk phosphorylation as a function of EGFR-mCitrine expression level as described in (D) (n=3 blots, *Figure 1—figure supplement 4*). All error bars correspond to standard error of the mean. EGFR, epidermal growth factor receptor.

The following figure supplements are available for figure 1:

**Figure supplement 1.** Relating ectopic EGFR-mCitrine expression with endogenous EGFR expression levels and dependency of autonomous EGFR activation on EGFR expression levels.

**Figure supplement 2.** Dependency of autonomous EGFR activation on EGFR expression levels.

**Figure supplement 3.** EGFR phosphorylation induced by PV-mediated PTP inhibition and EGF stimulation.

**Figure supplement 4.** Dependency of downstream EGFR signaling on EGFR expression levels.

To compare the level of spontaneous phosphorylation for the three sites, the fraction of auto-phosphorylated tyrosine relative to EGFR expression was normalized to the corresponding fraction after EGF stimulation (the maximally attainable phosphorylation at a given EGFR expression level). This 'ratio of fractions' (pre/EGF) gives a comparative measure of the auto-phosphorylation level of tyrosine residues as determined by immunofluorescence and Western blots. This analysis revealed that the extent of phosphorylation differed for the individual tyrosine residues, with a high correspondence between Western blot and immunofluorescence analysis (*Figure 1B,C*). In both experiments, the c-Cbl docking Y1045 site was significantly less phosphorylated as compared to the autocatalytic (Y845) and the signaling site (Y1068). In order to disentangle the contributions of EGFR kinase from PTP activity in generating this spontaneous phosphorylation profile, phosphorylation of EGFR was induced by pervanadate (PV)-mediated inhibition of PTPs (*Huyer et al., 1997*; *Haj, 2002*). In this way, the phosphorylation profile reflects the catalytic efficiency of EGFR kinase for the three tyrosine residues. The autocatalytic Y845 site was approximately 2.5 times more phosphorylated at 5' PV stimulation as compared to 5' EGF stimulation (*Figure 1D*). This rapid phosphorylation is consistent with a self-amplifying, autocatalytic phosphorylation. Both the Grb2 (Y1068) and c-Cbl (Y1045) docking sites were phosphorylated to a ~2.5 times lower level than Y845 (*Figure 1D*). The ratio of PV-induced over the spontaneous phosphorylation profiles (*Figure 1C*) allowed an estimation of the relative contribution of PTP activity in suppressing phosphorylation of the three tyrosines (*Figure 1E*). This analysis shows that the autocatalytic Y845 site requires higher PTP activity to be maintained in check as compared to the other two tyrosines. Moreover, the similar PTP activity that acts on Y1045 and Y1068 shows that the lower phosphorylation of Y1045 as compared to Y1068 is mostly due to differences in the catalytic efficiency of the EGFR kinase for these sites.

All three tyrosines exhibited a switch-like spontaneous phosphorylation response as function of EGFR expression (*Figure 1F*, *Figure 1—figure supplement 2–4*), indicating that the autocatalytic activation of EGFR is suppressed by PTP activity only up to a threshold of EGFR kinase activity. The phosphorylation of Y1045 was the least responsive to EGFR expression, which is consistent with a low catalytic efficiency of EGFR kinase for this site in the absence of ligand. Spontaneously phosphorylated EGFR largely remained at the PM (*Figure 1—figure supplement 1C*), which could result from the lack of c-Cbl binding to this inefficiently phosphorylated site. This steady state distribution is distinct from the EGF-induced internalized EGFR, which manifests in down-stream signaling where an increase in spontaneously phosphorylated EGFR clearly activates Akt but not Erk (*Figure 1G,H*, *Figure 1—figure supplement 4*). These results show that spontaneously and ligand-activated receptors behave as distinct signaling entities.

## Autonomously and ligand-activated EGFR differ in self-association

We investigated whether the distinct phosphorylation pattern and signaling behavior of spontaneously activated EGFR arises from a different self-association state as compared to ligand-bound

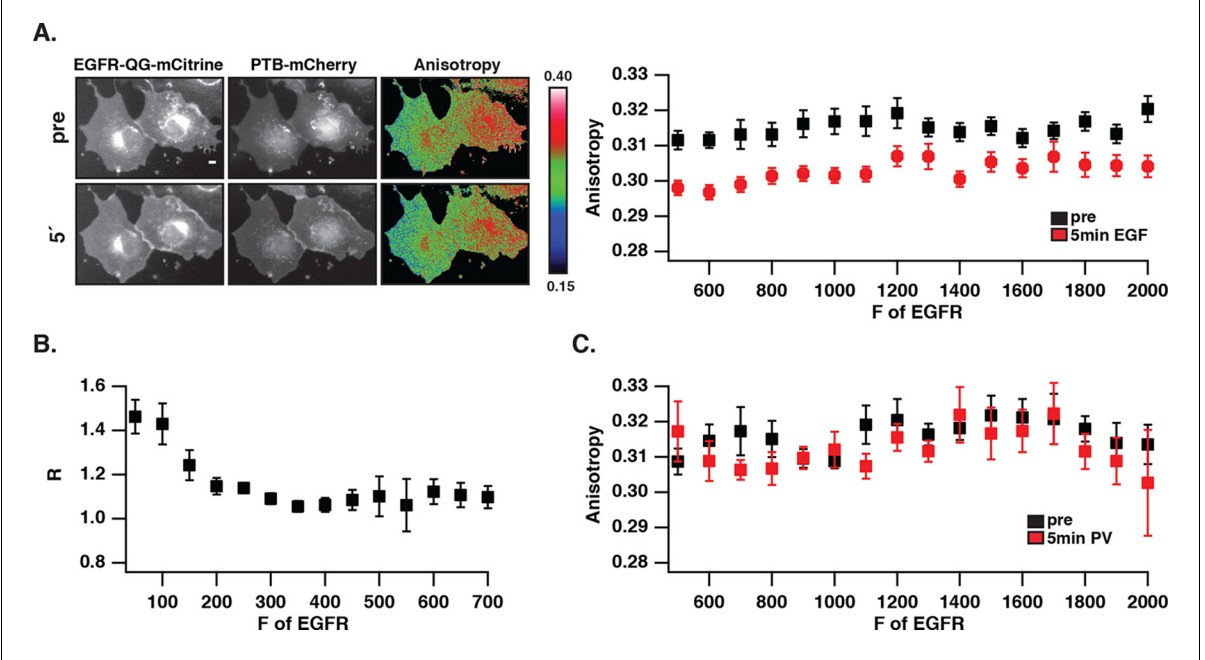

**Figure 2.** Autonomously and ligand-activated EGFR are different molecular states. (**A**) EGFR-QG-mCitrine anisotropy in COS-7 cells. Representative fluorescence images of EGFR-QG-mCitrine (first column), PTB-mCherry (second column), and anisotropy of EGFR-QG-mCitrine (third column) upon EGF stimulation for the indicated time. Scale bar: 10 µm. Graph shows the anisotropy of EGFR-QG-mCitrine versus its binned mean fluorescence intensity (F of EGFR) per pixel (black: pre-, red: 5-min post-stimulation with 100 ng/ml EGF). (**B**) Corresponding phosphorylation of EGFR-QG-mCitrine. Graph shows the recruitment (R, see 'Materials and methods') of PTB to EGFR versus the binned mean fluorescence intensity (F of EGFR) of EGFR-QG-mCitrine per pixel. (**C**) Anisotropy of spontaneously activated EGFR-QG-mCitrine upon PV-mediated PTP inhibition. Graph shows the anisotropy of EGFR-QG-mCitrine versus its binned mean fluorescence intensity (F of EGFR) per pixel (black: pre-, red: 5-min post-treatment with PV 0.33 mM; see *Figure 2—figure supplement 1B*). All error bars correspond to (standard error of the mean. EGFR, epidermal growth factor; PTB, phosphotyrosine-binding domain; PV, pervanadate.

The following figure supplement is available for figure 2:

**Figure supplement 1.** EGF-induced EGFR-QG-mCitrine phosphorylation and anisotropy upon PTP inhibition by PV.

receptors. To this end, fluorescence anisotropy microscopy was used to detect self-association of EGFR by homo-FRET (*Squire et al., 2004*; *Varma and Mayor, 1998*). For this purpose, an EGFR construct where mCitrine is inserted via a linker between amino acids Q958 and G959 (EGFR-QG-mCitrine) was generated that displayed a similar phosphorylation response to EGF as C-terminally tagged EGFR (EGFR-mCitrine) in COS-7 cells (*Figure 2—figure supplement 1A*). EGFR-QG-mCitrine exhibited homo-FRET upon dimerization as apparent from the EGF-induced decrease in anisotropy for all EGFR expression levels (*Figure 2A*). In order to assess EGFR phosphorylation in the same experiment, mCherry tagged PTB (PTB-mCherry) (*Batzer et al., 1995*; *Offterdinger et al., 2004*) that targets the phosphorylated signaling tyrosines 1148/1086 was co-expressed and its co-localization with EGFR-QG-mCitrine was determined at the cell periphery. At high EGFR-QG-mCitrine expression levels, co-localization with PTB-mCherry could not be enhanced by EGF stimulation, demonstrating the auto-phosphorylation of EGFR in the absence of ligand (*Figure 2B*). This spontaneously phosphorylated receptor remained mostly monomeric as shown by its high anisotropy (*Figure 2A*), even after PV-treatment that fully phosphorylates the receptor (*Figure 2C,1D*, *Figure 2—figure supplement 1B*). However, spontaneously activated EGFR at high expression levels still dimerizes upon ligand binding as apparent from the drop in anisotropy upon addition of EGF. These data are therefore consistent with mostly monomeric, spontaneously activated EGFR that is distinct from ligand-activated, self-associated EGFR.

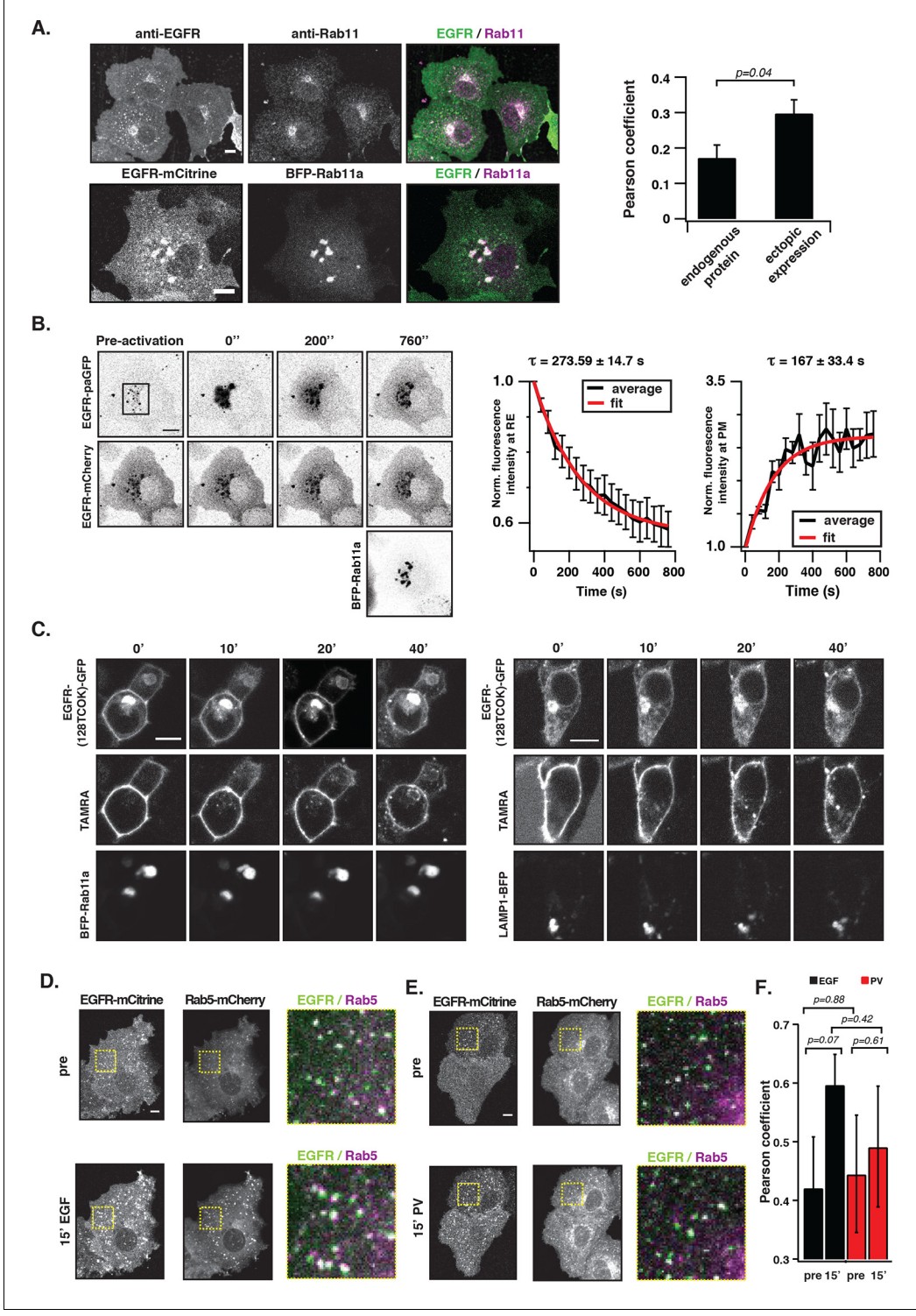

**Figure 3.** EGFR continuously recycles through the RE. (**A**) Co-localization of EGFR and Rab11. Left panel, first row: Immunostaining of endogenous EGFR (left), Rab11 (middle) and corresponding green/magenta overlay (right) in fixed COS-7 cells. Left panel, second row: Fluorescence images of EGFR-mCitrine (left), BFP-Rab11a (middle), and corresponding green/magenta overlay (right) in living COS-7 cells. Right panel: Quantification of co-localization between EGFR and Rab11 in A (n=15 cells), B (n=12) by Pearson's correlation coefficient. (**B**) Fluorescence redistribution after photoactivation of EGFR-paGFP on the recycling endosome. First row: EGFR-paGFP fluorescence at the indicated time in seconds (photoactivation area: black rectangle in the pre-activation image), second row: corresponding EGFR-mCherry fluorescence and image in third row: BFP-Rab11a fluorescence. Right graphs: Loss of EGFR-paGFP fluorescence at the RE normalized to EGFR-mCherry fluorescence (left) and corresponding gain of

*Figure 3. continued on next page*

*Figure 3. Continued*

EGFR-paGFP fluorescence at the PM (right), normalized to EGFR-mCherry fluorescence. Average trace (black ± SEM.; n=6 cells) was fitted to an exponential function (red) to retrieve time constants (τ) of EGFR fluorescence intensity lossat the RE and its gain at the PM. (C) Fluorescence redistribution of TAMRA-labeled-EGFR in HEK293 cells. Representative fluorescence images of EGFR-(128TCOK)-GFP (first row), TAMRA-labeled EGFR-(128TCOK)-GFP (second row), and BFP-Rab11a (third row, left panel) or LAMP1-BFP (third row, right panel) for the indicated time (min) in the respective columns (see 'Materials and methods'). (D) Co-localization of ligand-activated EGFR and Rab5 in COS-7 cells. Representative fluorescence images of EGFR-mCitrine (first column), Rab5-mCherry (second column), and green/magenta overlay of selected ROIs (third column), pre- and 15-min post-stimulation with EGF (100 ng/ml). (E) Co-localization of spontaneously activated EGFR and Rab5 in COS-7 cells. Representative fluorescence images of EGFR-mCitrine (first column), Rab5-mCherry (second column), and green/magenta overlay of selected ROIs (third column), pre- and 15-min post-treatment with 0.33 mM PV. (F) Quantification of co-localization between EGFR-mCitrine and Rab5-mCherry with Pearson's correlation coefficient upon EGF stimulation (100 ng/ml, n=5 cells) or PV treatment (0.33 mM, n=7 cells) for the indicated time points. All scale bars: 10 μm. EGFR, epidermal growth factor receptor; PM, plasma membrane; PV, pervanadate; RE, recycling endosome; SEM, standard error of the mean.

The following figure supplements are available for figure 3:

**Figure supplement 1.** Vesicular trafficking of autonomously and ligand-activated EGFR.

**Figure supplement 2.** Vesicular trafficking of autonomously-activated EGFR.

## Monomeric EGFR continuously recycles to the pericentriolar recycling endosome

To assess whether EGFR is maintained at the PM by vesicular recycling in COS-7 cells, we investigated if receptors partition in the Rab11-positive pericentriolar recycling endosome (RE) (*Ullrich, 1996*). Immunofluorescence staining showed co-localization of endogenous EGFR with Rab11 in unstimulated cells, which was also observed upon ectopic expression of EGFR-mCitrine and BFP-Rab11a (*Figure 3A*). However, ectopic expression of BFP-Rab11a enhanced the biogenesis of the RE (*Ullrich, 1996*), which shifted the distribution of EGFR to this compartment. Blocking protein synthesis with cycloheximide did not affect the apparent co-localization of Rab11 with EGFR showing that it did not originate from newly synthesized EGFR transiting through the Golgi via the secretory pathway (*Figure 3*).

Fluorescence loss after photoactivation (FLAP) and fluorescence recovery after photobleaching (FRAP) in the perinuclear area were performed to assess the extent of vesicular recycling of unliganded EGFR. Due to the shift in EGFR steady state distribution by ectopic Rab11a-BFP expression, most of the photo-activated or –bleached EGFR resided on the RE. The fluorescence loss after photoactivation of EGFR fused to photoactivatable GFP (EGFR-paGFP) (*Patterson, 2002*) on the RE (τ= 273 ± 15 s), and the concomitant gain at the PM (τ= 167 ± 33 s) shows that EGFR is recycled from the RE to the PM (*Figure 3B*). The difference in τ stems from the difference in surface area of the RE in relation to the PM, which determines the rate of RE-dissociation and PM-association. The residence time of EGFR at the RE was estimated to be ~7.6min ($k_{off}$ = 2.22 $10^{-3}$ s$^{-1}$) based on fitting the FLAP data to a two-compartment model (see 'Materials and methods'). From the fluorescence recovery after photobleaching EGFR-mCitrine on the RE (*Figure 3—figure supplement 1B*, τ= 540 ± 47 s), it was apparent that EGFR also enters the RE. The FLAP and FRAP data thus show that there is constitutive vesicular trafficking of EGFR to and from the RE. To ascertain that EGFR-mCitrine on the RE originates from the PM, a pulse/chase experiment was performed by conjugating tetrazine-TAMRA in a fast bioorthogonal chemical reaction to EGFR-GFP that contained a trans-cyclo-octene derivative of lysine (TCOK) (*Lang et al., 2012*). This unnatural amino acid was site-specifically and co-translationally incorporated by genetic code expansion in HEK293 cells into the extracellular part of EGFR at position 128 (*Lang et al., 2012*). TAMRA fluorescence of EGFR-(128TCOK)-GFP that was pulse labeled at the PM (*Figure 3C*, t=0 min) appeared 20 min later in Rab11a positive perinuclear endosomes, but not in LAMP1-BFP marked lysosomes (*Figure 3C*, *Figure 3—figure supplement 1C*). EGF stimulation, however, redirected a fraction of TAMRA-labeled EGFR-(128TCOK)-GFP to lysosomes (*Figure 3—figure supplement 1D*). This indicates that vesicular recycling between the PM and the RE maintains a steady state with the majority of EGFR at the PM, whereas EGF stimulation redirects self-associated EGFR towards lysosomes thereby depleting it from the PM.

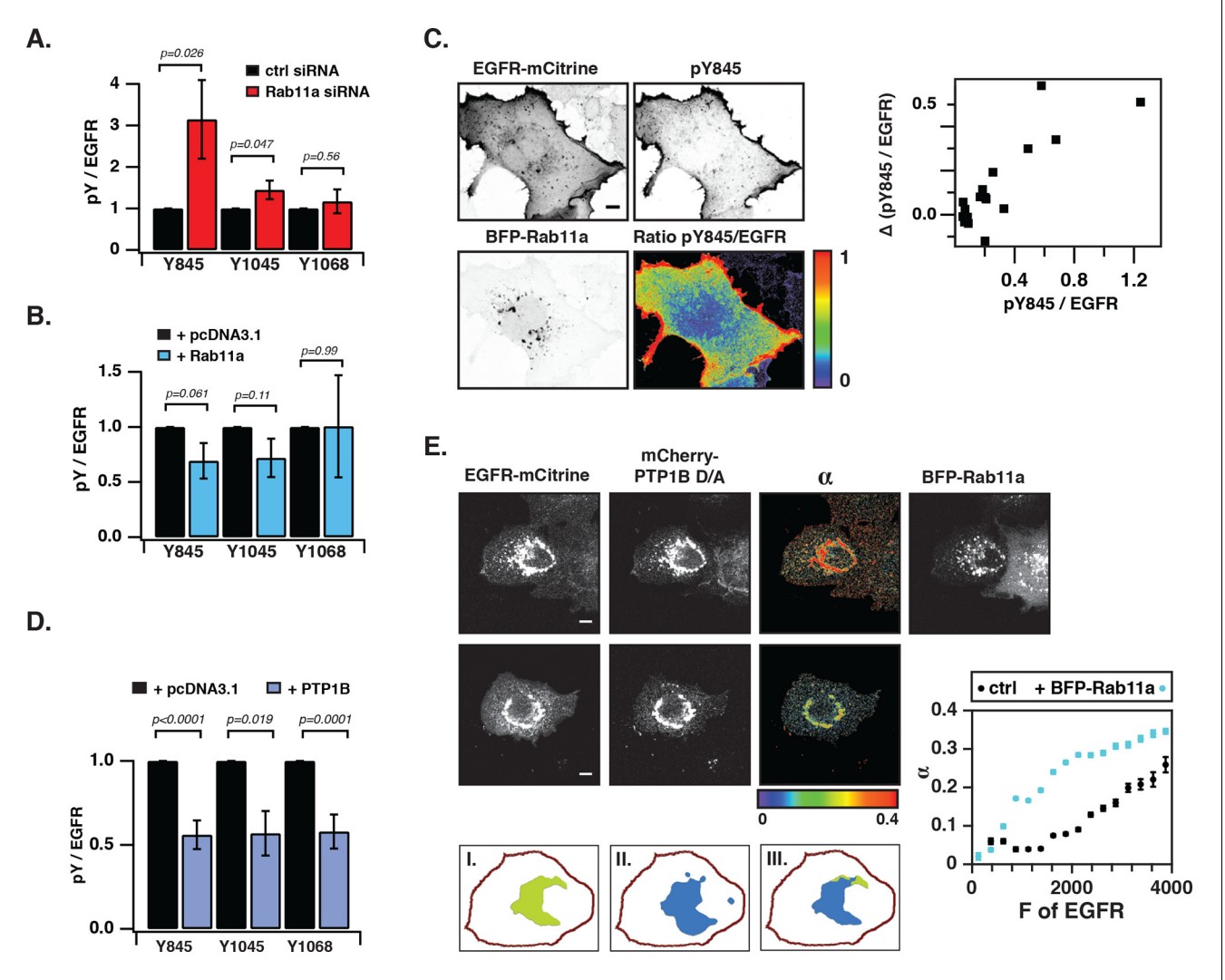

**Figure 4.** Suppression of spontaneous autocatalytic EGFR activation by vesicular recycling. (**A**) Effect of Rab11a knockdown on EGFR-mCitrine phosphorylation in COS-7 cells. The bar diagram shows the relative phosphorylation (pY/EGFR) of EGFR-mCitrine on Y845, Y1045, and Y1068 upon siRNA-mediated Rab11a knockdown normalized to pY/EGFR for cells transfected with non-targeting siRNA. For non-targeting siRNA: Y845 (n=69 cells), Y1045 (n=82), Y1068 (n=75). For Rab11a siRNA: Y845 (n=75 cells), Y1045 (n=74), Y1068 (n=83) (see *Figure 4—figure supplement 1A,B*). (**B**) Effect of BFP-Rab11a expression on EGFR-mCitrine phosphorylation in COS-7 cells. The bar diagram shows the relative phosphorylation (pY/EGFR) of Y845, Y1045 and Y1068 on EGFR-mCitrine upon ectopic expression of BFP-Rab11a normalized to pY/EGFR in the presence of empty pcDNA 3.1. For BFP-Rab11a ectopic expression: Y845 (n=52 cells), Y1045 (n=34), Y1068 (n=38) and for pcDNA 3.1: Y845 (n=125 cells), Y1045 (n=110), Y1068 (n=138) (see *Figure 4—figure supplement 2A*). (**C**) Spatial distribution of spontaneously phosphorylated Y845 in COS-7 cells. Left panel: Representative fluorescence images of EGFR-mCitrine (top left), immunostaining of pY845 (top right), BFP-Rab11a (bottom left), and a ratio image of pY845/EGFR-mCitrine (bottom right). Right panel: Graph shows the difference in Y845 auto-phosphorylation between the PM and RE (Δ pY845 over EGFR, see materials and methods) as a function of overall EGFR phosphorylation level in individual cells (pY845/EGFR). (**D**) Effect of ectopically expressed BFP-PTP1B on EGFR-mCitrine phosphorylation in COS-7 cells. The bar diagram shows the relative phosphorylation (pY/EGFR) of EGFR-mCitrine on Y845, Y1045, and Y1068 upon ectopic expression of BFP-PTP1B normalized to pY/EGFR upon transfection with empty pcDNA 3.1. For BFP-PTP1B ectopic expression: Y845 (n=69 cells), Y1045 (n=59), Y1068 (n=61) and for pcDNA 3.1: Y845 (n=54 cells), Y1045 (n=57), Y1068 (n=51) (see *Figure 4—figure supplement 2B*). (**E**) Spatial distribution of the interacting fraction (α, third column) of EGFR-mCitrine (first column) with mCherry-PTP1B D/A (second column) as detected by FLIM-FRET, with (upper row) or without (lower row) ectopic expression of BFP-Rab11a (4th column). Graph shows average α in regions of high EGFR-mCitrine intensity as a function of mean fluorescence (F of EGFR) with (blue, n=28 cells) or without ectopic expression of BFP-Rab11a (black, n=20). Lower row: percentage of EGFR/PTP1B D/A interactions in the vicinity of the RE was retrieved from the overlap (III) between areas with high α values (I) and areas with high intensity of BFP-Rab11a fluorescence (II) (see 'Materials and methods' and *Figure 4—figure supplement 3*) All scale bars: 10 μm. All error bars correspond to standard error of the mean. EGFR, epidermal growth factor receptor.

The following figure supplements are available for figure 4:

*Figure 4. continued on next page*

*Figure 4. Continued*

**Figure supplement 1.** Suppression of autonomous EGFR activation by recycling through perinuclear membranes.

**Figure supplement 2.** Suppression of autonomous EGFR activation by recycling through perinuclear membranes.

**Figure supplement 3.** Interaction of EGFR and PTP1B D181A in perinuclear areas.

We then investigated where this EGF-induced switch in vesicular trafficking occurs. Co-localization of EGFR-mCitrine with Rab5-mCherry in quiescent cells showed that unliganded EGFR is localized in EEs (*Figure 3D,E*, 0 min). Upon stimulation with EGF, the co-localized Rab5-mCherry and EGFR-mCitrine signals increased, confirming that liganded receptors transit via EEs to LE/Lysosomes and further increase the size and number of Rab5-mCherry-positive EEs (*Barbieri, 2000*) (*Figure 3D, F*). However, 15′ PV-mediated PTP inhibition only marginally affected the detectable amount and size of Rab5-mCherry positive EEs with phosphorylated EGFR (*Figure 3E,F*, *Figure 3—figure supplement 2A*). Photoactivation of EGFR-paGFP at the basal PM by TIRF microscopy, and the concurrent appearance of its fluorescence in Rab5-mCherry fluorescent vesicles confirmed that PM-residing EGFR recycles by transit through EEs (*Figure 3—figure supplement 2B*). These experiments show that recycling of monomeric EGFR via the EE compartment maintains its high steady-state distribution at the PM irrespective of phosphorylation state, whereas the different vesicular trafficking fates of unliganded versus liganded EGFR are decided at the level of the EE.

## Vesicular recycling suppresses spontaneous autocatalytic EGFR phosphorylation

We next investigated the coupling between EGFR auto-phosphorylation and vesicular recycling by decreasing/increasing the biogenesis of the pericentriolar RE. Decreasing RE biogenesis by knockdown of Rab11a with two independent siRNAs increased the phosphorylation of the autocatalytic Y845 ~ threefold and to a lesser extent the Y1045 (~1.5 fold), with no effect on EGFR-expression level (*Figure 4—figure supplement 1B*). However, no change in autophosphorylation for Y1068 was observed (*Figure 4A*, *Figure 4—figure supplement 1A–E*). Increasing RE biogenesis by overexpression of BFP-Rab11a gave the complementary result: phosphorylation of Y845 and Y1045 was lowered, whereas no change in phosphorylation was observed for Y1068 (*Figure 4B*, *Figure 4—figure supplement 2A*). The difference in Y845 phosphorylation between the PM and RE indicated that spontaneously activated EGFR is dephosphorylated on the RE (*Figure 4C*). These experiments are consistent with suppression of spontaneous EGFR phosphorylation at the perinuclear RE during recycling of the receptor.

The ER-associated, catalytically efficient PTP1B dephosphorylates internalized EGF-EGFR complexes mostly on perinuclear membranes (*Haj, 2002*; *Yudushkin et al., 2007*). The lowered phosphorylation of the three tyrosine sites upon ectopic expression of BFP-PTP1B showed that spontaneously phosphorylated EGFR is also dephosphorylated by PTP1B (*Figure 4D*, *Figure 4—figure supplement 2B*). To determine where this dephosphorylation occurs, we examined the interaction between EGFR-mCitrine and an mCherry-tagged substrate-trapping mutant of PTP1B (mCherry-PTP1B-D181A) by FLIM-FRET (*Haj, 2002*; *Sabet et al., 2015*). The spatially resolved fraction (α) of EGFR-mCitrine interacting with mCherry-PTP1B-D181A was computed using global analysis of FLIM data (*Verveer, 2000*; *Grecco et al., 2009*). An interacting fraction of EGFR-mCitrine and mCherry-PTP1B-D181A was only observed in the perinuclear region (*Figure 4E*). The mean spatial coincidence between this EGFR-mCitrine and mCherry-PTP1B-D181A interaction and BFP-Rab11a (63 ± 29% ; n= 20 cells; see 'Materials and methods') is consistent with EGFR being dephosphorylated by PTP1B on perinuclear membranes in the vicinity of the RE. BFP-Rab11a expression led to an elevated interacting fraction (α) at the RE due to the enhanced partitioning of EGFR in this compartment (*Figure 4E*, *Figure 4—figure supplement 3*). This shows that the steady state distribution of EGFR as determined by the anterograde and retrograde rates of recycling dictates the effectiveness of spontaneous phosphorylation suppression.

## Ubiquitin-mediated switching in vesicular trafficking of ligand-activated EGFR

Constitutive recycling of EGFR suppresses autonomous activation, but maintains a steady state distribution of EGFR at the PM irrespective of its activation state. In contrast, EGF stimulation leads to concomitant receptor depletion from the PM and its vesicular trafficking towards perinuclear endosomes (*Figure 5A*). At early times after stimulation, a transient translocation of EGFR-mCitrine to the PM and a parallel loss at the RE can be observed (*Figure 5B–D*). This transient shift in the steady-state distribution of EGFR likely occurs because recycling from the RE to the PM continues after EGF stimulation, whereas endocytosis of ligand-bound receptors at the PM is delayed.

We investigated the cause of this ligand-induced switch in receptor trafficking and how this affects its phosphorylation response. EGF stimulation induces EGFR ubiquitination by the E3-ligase c-Cbl (*Levkowitz et al., 1998*; *Waterman et al., 2002*) and we therefore asked whether c-Cbl-mediated ubiquitination is the key signal to differentiate trafficking of autonomously from ligand-activated EGFR. Ectopically expressed c-Cbl-mCherry was rapidly recruited to the EGF/EGFR complexes at the PM and subsequently observed on internalized EGFR-positive endosomes (*Figure 5A*). This ectopic expression strongly enhanced the rate of internalization after EGF stimulus (*Figure 5C*), which was concurrent with a diminished fraction of recycling EGF-EGFR (*Figure 5D*). EGF-stimulation led to ubiquitination and degradation of liganded receptors (*Figure 5E*, *Figure 5—figure supplement 1A*), whereas PV-mediated spontaneously activated EGFR exhibited much lower ubiquitination, was not degraded and did not alter its cyclic trafficking (*Figure 5E,F*). This shows that c-Cbl-mediated ubiquitination of EGFR causes a switch from recycling to unidirectional receptor trafficking toward the perinuclear cytoplasm. An EGFR mutant that is impaired in direct c-Cbl binding (EGFR-Y1045F-mCitrine) (*Grøvdal et al., 2004*) further corroborated this conclusion by continuing to recycle via the RE after ligand stimulation, thereby preventing its degradation (*Figure 5G,H*, *Figure 5—figure supplement 3A–C*). This constitutive recycling also resulted in an equally sustained phosphorylation in response to EGF, as compared to the adaptive phosphorylation response of wild type EGFR (*Figure 5I*, *Figure 5—figure supplement 3B*). The same stabilizing features were additionally observed for an EGFR mutant (EGFR-Y1045F/Y1068F/Y1086F-mCitrine) that is impaired in both direct and indirect c-Cbl binding via Grb2 (*Figure 5H,I*, *Figure 5—figure supplement 3A–C*). This shows that the EGF-mediated ubiquitin-switch in vesicular traffic that imposes a finite EGFR signal duration is mediated by c-Cbl binding to phosphorylated Y1045. However, PV-mediated spontaneous EGFR activation caused a similar or higher phosphorylation of Y1045 without c-Cbl recruitment (*Figure 5—figure supplement 2*), leading to inefficient receptor ubiquitination (*Figure 5E*, *Figure 5—figure supplement 1B*). This indicates that phosphorylation of Y1045 alone is not sufficient for EGFR ubiquitination by c-Cbl and that EGF-induced receptor association is an essential requirement. EGF-induced ubiquitination thus is a secondary signal that redirects EGFR toward perinuclear areas with high PTP1B activity to inactivate signaling of the liganded receptor (*Figure 5—figure supplement 4*) and further commit it for lysosomal degradation to avoid reactivation at the PM.

## Elucidating the systemic rationale for the switch in vesicular EGFR trafficking

In order to better understand why vesicular EGFR trafficking switches from recycling to unidirectional trafficking upon EGF stimulation, we incorporated our experimental findings and previous knowledge into a minimal model in which the effect of spatial partitioning of phosphates and EGFR trafficking on its activity could be investigated. The set of reactions in this model (*Figure 6A*) encompassed the following precepts:

1. Autonomous phosphorylation of unliganded EGFR occurs spontaneously (*Figure 1A*) in trans via short-lived dimers when one EGFR molecule attains an active conformation through thermal fluctuations (*Figure 6A*, *Equation 1*). This notion is supported by the independence of anisotropy on EGFR expression level (*Figure 2A*). This second order autonomous phosphorylation reaction (*Figure 6A*, *Equation 2*) occurs with very slow kinetics due to the low fraction of EGFR in the active conformation. The balance of phosphorylation and de-phosphorylation (*Figure 6A*, *Equation 3*) determines the fraction of phospho-EGFR (*Figure 6B*). Neither the absolute value of phosphatase and kinase rate constant, nor whether these are mass-action or Michaelis-Menten reactions is relevant, as we monitor the resulting phosphorylation state.

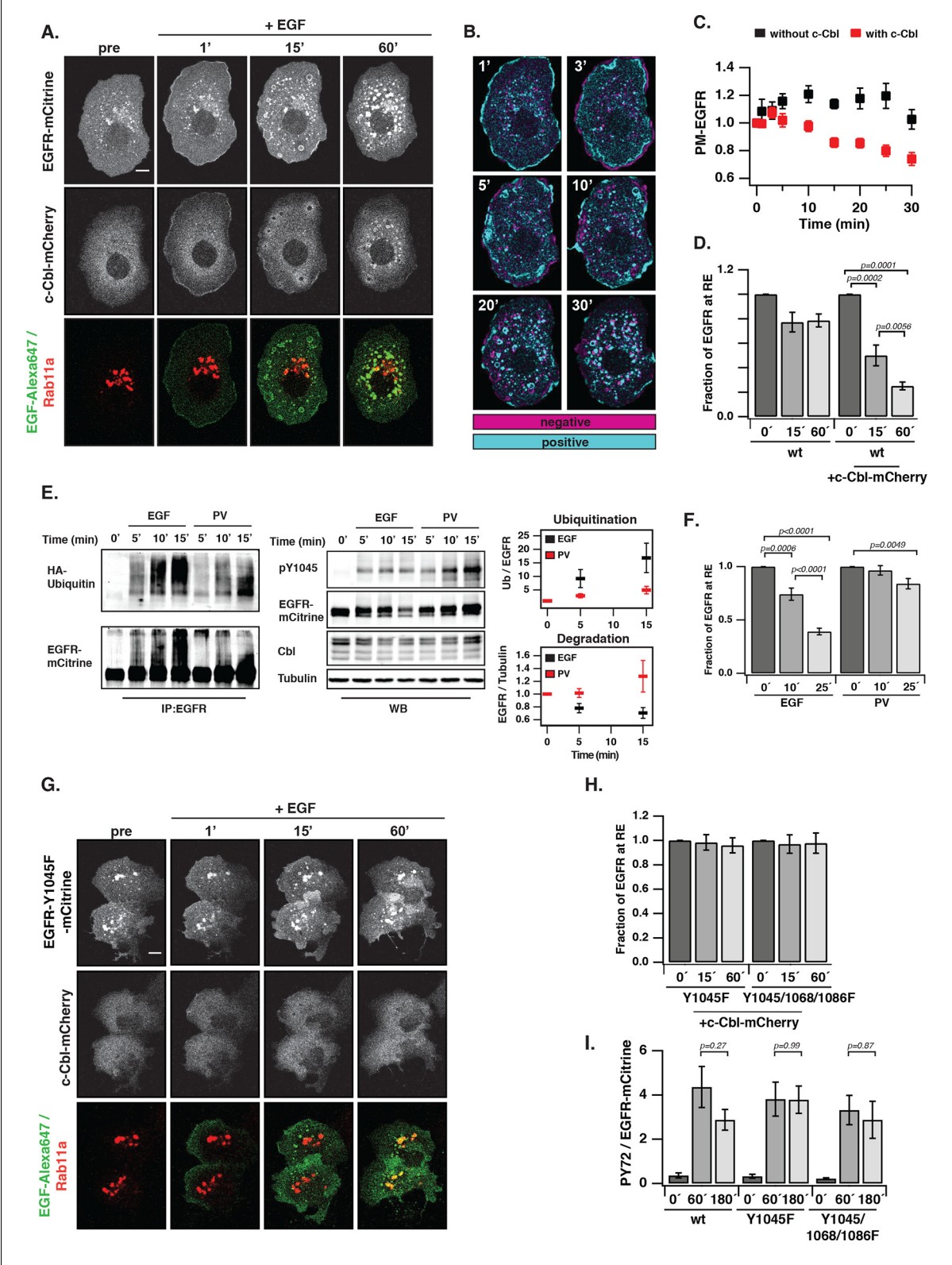

**Figure 5.** Ubiquitin-mediated switch in ligand-activated EGFR trafficking. (**A**) Time-lapse of EGFR and Rab11a co-localization after EGF stimulation of COS-7 cells with Alexa647-labeled EGF (5 ng/ml). Representative fluorescence images of EGFR-mCitrine (first row), c-Cbl-mCherry (second row), and BFP-Rab11a merge with Alexa647-labeled EGF (third row) at the indicated time in minutes. (**B**) Change in spatial distribution of EGFR-mCitrine upon EGF stimulation calculated as the difference between each time point and the one acquired prior to it, indicates areas with increased (cyan) or

*Figure 5. continued on next page*

*Figure 5. Continued*

decreased (magenta) fluorescence. (**C**) Plot shows the normalized average fraction ± SEM of EGFR-mCitrine at the PM (PM-EGFR over total EGFR) in the presence (red, n=10 cells) or absence (black, n=6 cells) of c-Cbl-mCherry over time (see 'Materials and methods'). (**D**) Fraction of EGFR-mCitrine fluorescence at the RE with (n=10 cells) and without (n=6 cells) ectopic expression of c-Cbl-mCherry upon EGF stimulation for the indicated time. (**E**) Differential ubiquitination of ligand- and autonomously-activated EGFR. COS–7 lysates immunoprecipitated with anti-EGFR (left panel) or blotted for total proteins (middle panel). IP was probed with anti-HA (HA-ubiquitin) and anti-GFP (EGFR-mCitrine) and total lysates were probed with anti-GFP (EGFR-mCitrine), anti-Cbl (c-Cbl-mCherry), anti-tubulin (Tubulin), and anti-pY1045. Right panel: Quantification shows relative EGFR ubiquitination (Ub over EGFR) in the upper graph and degradation (EGFR over Tubulin) in the lower graph upon EGF stimulation (100 ng/ml) or PV treatment (0.33 mM) for the indicated time in minutes (see *Figure 5—figure supplement 1A,B*). (**F**) Fraction of EGFR-mCitrine fluorescence at the RE upon stimulation with 100 ng/ml EGF (n=12 cells) or 0.33 mM PV (n=14 cells) for the indicated time (see *Figure 5—figure supplement 2*). (**G**) Time-lapse of EGFR-Y1045F-mCitrine and BFP-Rab11a co-localization after EGF stimulation of COS-7 cells with Alexa647-labeled EGF (5 ng/ml). Representative fluorescence images of EGFR-Y1045F-mCitrine (first row), c-Cbl-mCherry (second row), and BFP-Rab11a merge with Alexa647-labeled EGF (third row) at the indicated time in minutes. (**H**) Fraction of EGFR-Y1045F-mCitrine (n=9 cells) or EGFR-Y1045/1068/1086F-mCitrine (n=9 cells) fluorescence at the RE upon EGF stimulation for the indicated time (min) in the presence of ectopically expressed c-Cbl-mCherry (see *Figure 5—figure supplement 3A*). (**I**) Sustained phosphorylation of EGFR-Y1045F-mCitrine and EGFR-Y1045/1068/1086F-mCitrine as compared to EGFR-mCitrine upon EGF stimulation. COS-7 lysates were probed with generic phosphotyrosine (PY72) and anti-GFP (EGFR-mCitrine) and bar diagram shows relative EGFR phosphorylation (PY72/EGFR) upon EGF stimulation (100 ng/ml) for the indicated time (see *Figure 5—figure supplement 3B*). All blots are n=3 (mean ± SEM). All scale bars: 10 μm. EGFR, epidermal growth factor receptor; SEM, standard error of the mean.

The following figure supplements are available for figure 5:

**Figure supplement 1.** EGF-induced vesicular trafficking of wild-type EGFR-mCitrine and differential ubiquitination of autonomously and ligand-activated EGFR.

**Figure supplement 2.** Vesicular trafficking of autonomously versus ligand-activated EGFR to the RE.

**Figure supplement 3.** EGF-induced vesicular trafficking of EGFR-Y1045/Y1068F/Y1086F-mCitrine and c-Cbl-mediated degradation profiles of EGFR upon EGF stimulation.

**Figure supplement 4.** Interaction of EGFR and PTP1B D181A in perinuclear areas upon EGF stimulation.

Even time units are arbitrary, but can be used to relate different scenarios of kinase activity and PTP partitioning.

2. Phosphorylation of tyrosine 845 introduces an autocatalytic feedback (*Figure 6A*, *Equation 4*, *Figure 1A*), where we assume that EGFR has the same kinase activity as the spontaneously attained active conformation.

3. Endocytosis and recycling creates a steady state partitioning (*Figure 3*) between PM-bound and internalized EGFR (*Figure 6A*, *Equation 5*).

4. Phosphatase activity is partitioned between both compartments. Combined with recycling of EGFR, this represents an additional effective phosphatase activity at the PM. We investigated the effect of differential partitioning of phosphatases as well as the steady state distribution of EGFR on phosphorylation of the autocatalytic site (Y845).

5. EGF binding to EGFR maximizes its rate of phosphorylation and maintains its kinase fully active irrespective of its phosphorylation state (*Figure 6A*, *Equation 6*, *Figure 1A*). Only high phosphatase activity can effectively dephosphorylate liganded EGFR in order to negate its signaling activity (*Figure 5—figure supplement 4*).

In a system where autonomous phosphorylation of EGFR results purely from leaky kinase activity due to conformational fluctuations, low phosphatase activity suffices to prevent all EGFR from eventually being phosphorylated. We normalized the phosphatase rate constant to a unit 1 as that phosphatase activity, which allows an autonomous phosphorylation of 50% EGFR in steady state. This phosphatase activity balances the 1% of all EGFR spontaneously attaining an active conformation. Under these conditions, a $k_{PTP}$ = 15 units suppresses phosphorylation to a level below 10% (*Figure 6B*).

Introducing autocatalysis results in a massive acceleration of phosphorylation. If phosphatase activity was absent in this system, all EGFR would be phosphorylated 250 times more rapidly than without the autocatalytic feedback (compare *Figure 6C*, dashed curve with *Figure 6B*, green curves). In combination with the low phosphatase activity of 15 that maintains autonomous

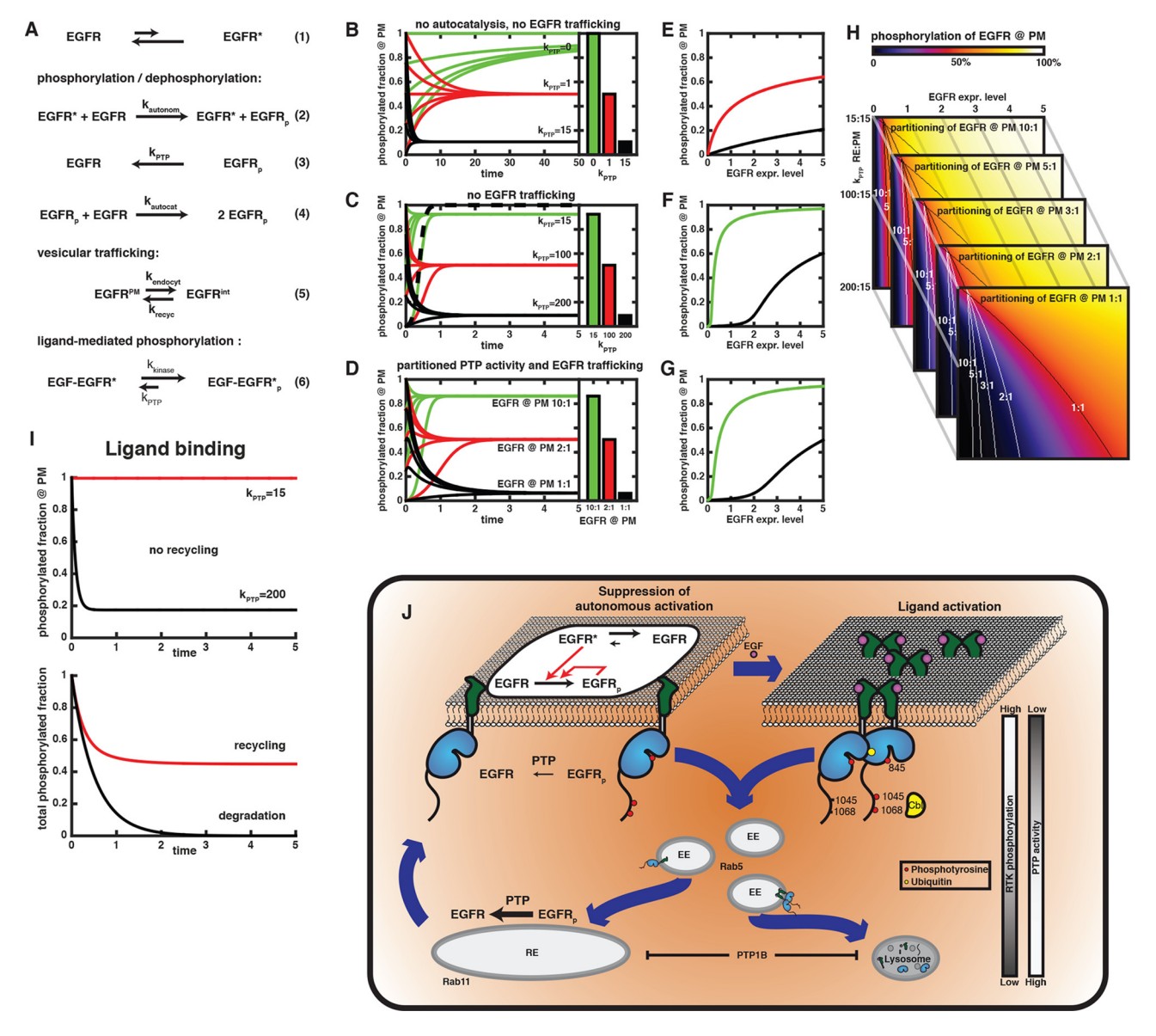

**Figure 6.** Compartmental model for EGFR spontaneous autocatalytic activation supression. (**A**) The set of conversion reactions based on which ordinary differential equations were numerically integrated in Mathematica (see 'Materials and methods'). EGFR fluctuates between an inactive and an active conformation – denoted by star (1). The active conformation phosphorylates EGFR with the rate constant $k_{autonom}$ that encompasses the fraction of active conformation (2). Here, only phosphorylation of an autocatalytic tyrosine is considered. Phosphorylated EGFR is dephosphorylated ($k_{PTP}$; 3) and can *auto*catalytically phosphorylate EGFR ($k_{autocat}$; 4) similar to (2). All species of EGFR exchange between two compartments (from PM to internal – $k_{endocyt}$ – and back – $k_{recyc}$; 5). $k_{PTP}$ may vary between both compartments. Ligand-binding locks EGFR in the active conformation, irrespective of its phosphorylation state ($k_{kinase}$ over $k_{PTP}$). (**B**—**D**) Time traces of EGFR phosphorylation and bar diagrams of the corresponding phospho-EGFR steady states at the PM for different parameters (green – phospho-EGFR >80%, red– phospho-EGFR ~50%, black – phospho-EGFR <10%): (**B**) without autocatalytic feedback or recycling and increasing values of $k_{PTP}$; (**C**) with autocatalysis, but no recycling and increasing values of $k_{PTP}$; and (**D**) autocatalysis, recycling and $k_{PTP}$=15 at the PM versus $k_{PTP}$=200 for the internal EGFR fraction with increasing steady state partitioning of EGFR towards the internal fraction. Dashed line in (**C**) is in the absence of phosphatase activity (phospho-EGFR rapidly approaches 100%). (**E**—**G**) Steady state phosphorylation of EGFR at the PM for the situations displayed in (**B**–**D**) upon varying the total amount of EGFR per cell (expression level). Colors (green, red, black) correspond to the parameters in (**B**–**D**). (**H**) 2D plots of EGFR phosphorylation (color code displayed on top) analogous to the last line plot in (**E**) for different expression levels of EGFR (x-axis) versus increasing phosphatase activity acting on the internal EGFR fraction (y-axis). Each image corresponds to a different steady state partitioning of EGFR towards the PM (10:1, 5:1, 3:1, 2:1, 1:1); lines in each image mark 50% EGFR phosphorylation (red color) for the corresponding steady state EGFR partitioning. (**I**) Simulation of EGF-induced activation of EGFR. In the absence of recycling (upper time traces), zeroth order phosphorylation rate of liganded EGFR locked in the active conformation results in a stable phosphorylation

*Figure 6. continued on next page*

*Figure 6. Continued*

of all EGFR for low PTP activity (red curve) or an extremely rapid dephosphorylation for high PTP activity (black curve). Recycling in the presence of partitioned PTP activity (as in (D), $k_{PTP}$=15 at the PM and $k_{PTP}$=200 on endomembranes) results in 100% phosphorylation at the PM and low phosphorylation of EGFR on endomembranes. The total phosphorylation of EGFR in the cell (abscissa) is then determined by steady state EGFR partitioning (~50% for equipartitioning, red curve). Only degradation of the internal fraction with ongoing endocytosis allows finite signaling whose duration is determined by the kinetics of endocytosis (black curve). (J) Schematic of switch in EGFR vesicular traffic. Vesicular recycling (thick blue arrows) of EGFR monomers from the PM through the pericentriolar RE within areas of high PTP activity and back continuously suppresses autonomous receptor phosphorylation. Horizontal black arrows: Chemical conversions, vertical red arrows: Causalities (left). Following EGF-binding to the receptor, fully active and phosphorylated dimers share the same entry into early endosomes as the monomer. However, ligand-induced ubiquitination of EGFR clusters mediates a switch in the endocytic trafficking from continuous recycling to unidirectional trafficking. By depleting the pool of monomeric EGFR at the RE, this results in a transient translocation of the receptors to the PM reinforcing activation, and a subsequent depletion from the PM by delayed endocytosis. During endosomal transit from the PM to LEs, receptors signal in the cytoplasm until their dephosphorylation in perinuclear areas with high PTP activity (right) and subsequent degradation in lysosomes.

phosphorylation below 10% (black curves in *Figure 6B*), autocatalysis causes a highly phosphorylated fraction. In this case, a phosphatase activity of 200 is required to suppress autocatalytic phosphorylation below 10% (*Figure 6C*). However, this level of phosphatase activity rapidly suppresses phosphorylation of ligand-bound EGF-EGFR (*Figure 6I*, upper graph).

We then posed the question, whether steady state EGFR distribution caused by recycling and a partitioning of phosphatase activity between PM and RE can counteract autocatalytic phosphorylation while maintaining responsiveness of EGFR to EGF. We therefore started from a condition where EGFR at the PM is subjected to low phosphatase activity of ($k_{ptp}$ = 15) that is sufficient to prevent autonomous phosphorylation without suppressing phosphorylation of liganded EGFR, nor autocatalytic phosphorylation (compare *Figure 6B,C*). Additionally, a PTP activity that is sufficient to counter autocatalytic phosphorylation ($k_{ptp}$ = 200) was assigned to intracellular membranes. Upon exchange of EGFR between both compartments by recycling, the relative amount of EGFR at the PM determines its phosphorylation level (*Figure 6D*). EGFR phosphorylation is high at a 10:1 steady state distribution, while equipartitioning of EGFR leads to a phosphorylation well below 10%. Vesicular recycling thus acts as an additional, effective phosphatase activity that can suppress autocatalytic phosphorylation.

Why is the system using recycling in order to generate an effective phosphatase activity at the PM? Since autocatalytic activity is a non-linear process that is strongly dependent on the density of EGFR at the PM, we also compared the phosphorylation of EGFR as a function of EGFR expression level in the three cases described above: no autocatalysis, no recycling, recycling, and partitioning of phosphatase activity (*Figure 6E–G*). Without autocatalytic phosphorylation, the second-order rate of autonomous phosphorylation causes a square-root dependence of phosphorylation on EGFR expression. For the moderate phosphatase activity of 15 that suppresses autonomous phosphorylation, this dependence is stretched out and it appears almost linear (black curve in *Figure 6E*). However, this amount of phosphatase activity combined with autocatalytic phosphorylation gives rise to a low threshold (< 0.1) in EGFR expression, beyond which phosphorylation increases sharply as a function of EGFR levels. This threshold increases with phosphatase activity (expression level of 2 for a phosphatase activity of 200, compare *Figure 6F*). In the case of partitioned phosphatase activity, the dependence of EGFR phosphorylation on expression is similar to the scenario without recycling (*Figure 6G*). This shows that the system of partitioning phosphatase activity and EGFR trafficking poses a valid alternative to a purely PM-based solution of keeping autocatalytic phosphorylation in check. The essential difference is that a partitioned phosphatase activity achieves this while maintaining a low phosphatase activity at the PM, where ligand-mediated activation of EGFR occurs.

So far, we exemplary considered a partitioning of phosphatase activities of 15:200 between PM and endomembranes. We find that the autocatalytic threshold (black and white lines in *Figure 6H*) depends on this phosphatase partitioning, as well as on the EGFR distribution. Even if there is 10 times more EGFR on the PM, strongly asymmetric phosphatase partitioning hardly shifts the threshold. However, for more equal distributions of EGFR (>2:1), a concurrent asymmetric phosphatase partitioning provides a large shift in the threshold. This corresponds to a safeguard against autocatalytic phosphorylation even at high EGFR expression. A low phosphatase activity at the PM suffices in combination with the suppressive recycling to maintain the level of EGFR phosphorylation below the

autocatalytic threshold. For liganded EGFR, vesicular recycling leads to efficient dephosphorylation on perinuclear membranes with their high local phosphatase activity. However, the stabilization of the active EGFR conformation by the ligand causes its kinase activity to surpass suppression by low local phosphatase activity once liganded EGFR recycles to the PM (*Figure 6I*, lower graph, red curve). This is in stark contrast to unliganded dephosphorylated EGFR returning from the perinuclear area that requires slow autonomous phosphorylation as the trigger to return to the active state. The slow rate of recycling thus proves ineffective to maintain ligand-activated EGFR in a dephosphorylated state, while it can suppress autonomous phosphorylation (compare *Figure 6D* with *Figure 6I*, lower graph). The system is therefore left with one way to shut down signaling after ligand stimulation: reducing the amount of EGF-EGFR complexes in the whole cell by lysosomal degradation to reset the kinase-phosphatase balance to a level below the autocatalytic threshold. The scheme in *Figure 6J* summarizes this switch from continuous suppressive EGFR recycling to directional trafficking leading to finite signaling

## Discussion

Ligand-induced EGFR signaling is involved in numerous cellular responses in health and disease, with a documented link between aberrant ligand-independent EGFR activation through overexpression or mutations and tumorigenesis (*Arteaga and Engelman, 2014*). It remains unclear how spontaneous auto-activation of EGFR is suppressed, while still maintaining responsiveness to physiological stimuli. Here, we show that a switch from cyclic to unidirectional vesicular traffic of EGFR enables its interaction with spatially organized PTPs to suppress spontaneous auto-activation while allowing for ligand-induced signal propagation. Otherwise, suppression of spontaneous autocatalytic activation at the PM would be incompatible with ligand-induced signaling of EGFR. Partitioning high PTP activity away from the PM allows these opposing requirements, because of two modes of spontaneous EGFR phosphorylation: 1) slow autonomous phosphorylation originating from a low fraction of EGFR that is in the active conformation due to thermal fluctuations; and 2) rapid autocatalytic phosphorylation due to phosphorylation of the autocatalytic site Y845. Low PTP activity at the PM is sufficient to suppress autonomous EGFR phosphorylation and can be overcome by ligand-activated EGFR. In contrast, the extremely fast kinetics of autocatalytic phosphorylation can only be suppressed by PTPs with high catalytic efficiency. This would hinder EGF-induced phosphorylation of EGFR at the PM and thereby prevent its signaling. This is the reason why shutdown of EGFR signaling occurs after its internalization by vesicular trafficking to the perinuclear area, where high PTP activity dephosphorylates the liganded receptor without interfering with EGF-responsiveness at the PM. This feature of the perinuclear area is also capitalized by monomeric, spontaneously phosphorylated EGFR through continuous vesicular trafficking into the same spatial location. The two different 'species' of phosphorylated EGFR – —spontaneously and ligand-activated – —are distinguished by their ubiquitination that depends on phosphorylation of tyrosine 1045 and self-association of EGFR. EGF-EGFR complexes travel in a unidirectional way to this perinuclear area due to their ubiquitination and will not leave it again, whereas spontaneously phosphorylated EGFR will recycle to the PM after dephosphorylation at the RE. Slow spontaneous autonomous receptor phosphorylation will then start again at the PM. However, before the threshold is reached, where rapid autocatalysis would accelerate phosphorylation of Y845, trafficking to the RE restarts the suppressive cycle. The primary function of EGFR recycling is therefore to suppress spontaneous phosphorylation of Y845 that leads to autocatalysis. This notion is consistent with our finding that – —out of the three investigated tyrosines – —phosphorylation of the autocatalytic Y845 site was the most efficiently suppressed by vesicular recycling of EGFR (*Figure 4A*). This has implications for spontaneously versus EGF-activated EGFR- signaling. Since auto-activated EGFR is maintained at the PM, it can activate Akt via PIP3 production by PI3K at the PM (*Cantley, 2002*), resulting in survival signaling. In contrast, we observed unresponsiveness of Erk to spontaneous auto-activation of EGFR. Possible reasons for this could be that negative feedback in the Erk circuitry (*Fritsche-Guenther et al., 2011*) results in an adaptive response to sustained EGFR activity at the PM or that efficient activation of Erk only occurs upon endocytosis of the receptor. In general, our data show that interference with vesicular trafficking by knock down of Rab proteins can have substantial pleiotropic effects on growth factor receptor signaling activity.

We have shown here that the ER-associated PTP1B efficiently dephosphorylates the autocatalytic Y845. The highest intrinsic activity of PTP1B has been demonstrated to be on perinuclear membranes (*Yudushkin et al., 2007*), whereas only a small fraction of its activity can reach the PM at points of cell–cell contacts (*Haj, 2002*). In addition to PTP1B, TCPTP is also spatially segregated from PM-localized EGFR by its association with the ER and has been shown to dephosphorylate Y845 (*Mattila et al., 2005*). Thus, the localization of these ER-anchored PTPs with a high catalytic efficiency toward phosphorylated Y845 in the perinuclear area is compatible with the presented model of maintaining unliganded EGFR phosphorylation at the PM below the autocatalytic threshold. In this way, even a marginal PTP activity at the PM suffices to silence the signaling tyrosines. In this regard, several RPTPs (i.e. RPTPJ, G, A) that dephosphorylate EGFR at the PM (*Rotin et al., 1992*; *Tarcic et al., 2009*; *Leung Cheung et al., 2015*) exhibit a high preference for the signaling tyrosines Y1068 and Y1086 (*Leung Cheung et al., 2015*; *Tarcic et al., 2009*). In addition, the catalytic efficiency of RPTPA for phosphorylated peptide substrates is three orders of magnitude lower than that of PTP1B or TC-PTP (*Selner et al., 2014*). This agrees with our model where low PTP activity at the PM can only suppress slow autonomous phosphorylation in case that high perinuclear PTP activity suppresses autocatalytic activation. This is substantiated by the fact that inhibiting internalization of EGFR results in its prolonged activity at the PM (*Offterdinger and Bastiaens, 2008*), whereas inhibition of EGFR kinase activity with an ATP-binding site competitor results in rapid dephosphorylation of signaling tyrosines (*Offterdinger et al., 2004*). In the latter case, there is no autocatalytic maintenance of the kinase activity and the PM-resident PTP activity suffices to dephosphorylate EGFR on signaling tyrosines.

We argue that the unidirectional vesicular trafficking of liganded receptors to the perinuclear area determines the duration of EGFR signaling in response to EGF. During vesicular transit of EGF-EGFR complexes to perinuclear areas with the highest PTP1B activity (*Yudushkin et al., 2007*), growth factor signals are propagated in the cytoplasm (*Lai, 1989*; *Di Guglielmo et al., 1994*; *Wouters and Bastiaens, 1999*; *Sorkina et al., 1999*), indicating a link between vesicular trafficking and duration of EGFR signaling response. Such a response typically occurs on the tens-of-minutes time-scale, corresponding to characteristic vesicular trafficking rates. This time-scale of EGFR response cannot be achieved by cytosolic, regulated PTPs (i.e. SHP1/2) that couple to EGFR phosphorylation in a negative feedback manner (*Grecco et al., 2011*). The recruitment of these diffusible PTPs to the receptor on the time scale of seconds would dephosphorylate EGFR too rapidly to signal on a time scale of minutes. By the same argument, suppression of spontaneous autocatalytic activation at the PM by regulated PTPs is incompatible with prolonged EGFR signaling.

Both, EGF-EGFR complexes as well as recycling receptors transit via Rab5 positive EEs, into the LE and RE, respectively (*Figure 3D,E*). This suggests that sorting of monomeric EGFR to the RE and self-associated, ubiquitinated EGFR to the LE happens at the early endosomal compartment. Why is it necessary to re-route ligand-activated receptors by endosomal sorting away from the unliganded EGFR that keeps recycling? The answer to this question lies in the stability of the ligand-receptor complexes. Liganded EGFR maintains an active conformation even after dephosphorylation in the perinuclear area, and therefore will be rapidly re-phosphorylated in the environment of low PTP activity at the PM. This would sustain signaling as it has been observed for the c-Cbl-binding-impaired mutants, which continued recycling upon EGF stimulation (*Figure 4B–D*). Importantly, this reactivation does not occur for spontaneously activated EGFR because dephosphorylation of Y845 inactivates the conformation that results in rapid autocatalytic phosphorylation. The protective recycling can be exploited by oncogenic versions of EGFR (*Chung et al., 2009*) that show impaired c-Cbl association (*Shtiegman et al., 2007*), or c-Cbl mutations that abolish interaction with EGFR, or within cells with altered c-Cbl activity, all leading to insufficient EGFR ubiquitination, subsequent recycling and sustained downstream signaling (*Thien and Langdon, 1997*; *Peschard and Park, 2003*; *Ravid et al., 2004*). Recycling also occurs at low doses of EGF stimulation (*Sigismund et al., 2005*), where some receptors might autocatalytically activate but fail to be ubiquitinated due to lack of stable EGF-induced self-association. However, as shown in this study, c-Cbl is a limiting factor in rerouting EGF-EGFR complexes to lysosomes under saturating doses of EGF. Therefore, the level of c-Cbl expression in different cells might dictate the fraction of EGF-EGFR complexes that keep recycling.

Even though re-routing of liganded EGFR occurs on EEs, EGF stimulation leads to an apparent transient translocation of EGFR to the PM, followed by its depletion (*Figure 5A–C*). This shows that

EGF-induced receptor clustering or ubiquitination results in a delayed internalization, where continued trafficking from the RE results in this perceived net translocation. The delayed internalization might point at a different internalization and trafficking route to EEs of monomeric receptors as compared to ubiquitinated EGF-EGFR complexes (*Sigismund et al., 2013*).

One of the decisive factors for this differential sorting of spontaneously and ligand-activated receptors is phosphorylation of the c-Cbl binding site Y1045. We could show that - —out of the three investigated tyrosine residues - —it was the least efficiently phosphorylated by spontaneous EGFR kinase activity (*Figure 1B–D*). This is in accordance with data showing that Y1045 is inefficiently auto-phosphorylated under oxidative stress, in contrast to Y845 (*Ravid et al., 2002*). The fact that monomeric EGFR was inefficiently ubiquitinated upon PTP inhibition (*Figure 5E*), but displayed a similar phosphorylation on Y1045 as compared to EGF stimulation, suggests that rather than Y1045 phosphorylation alone, the formation of EGFR oligomers is a decisive factor for c-Cbl binding and efficient ubiquitination.

The distinct molecular states of spontaneous activated versus ligand-activated EGFR that are recognized and processed differently by the endocytic machinery, most likely evolved to provide robustness in ligand responsiveness against the gain in plasticity from its autocatalytic activation mechanism. This feature therefore seems to be common for other receptor tyrosine kinases as we have shown for the cell guidance ephrin receptor type-A2 (EphA2) (*Sabet et al., 2015*).

# Material and methods

## Antibodies

### Primary antibodies

Mouse anti-HA (MMS-101P, Biolegend, San Diego, CA), mouse anti-phospho tyrosine (PY72) (P172.1, InVivo Biotech Services, Henningsdorf, Germany), rabbit anti-c-Cbl (sc-170, Santa Cruz Biotechnologies, Heidelberg, Germany), mouse monoclonal anti-α-Tubulin (Sigma Aldrich, St. Louis, MO); living colors rabbit anti-GFP (632593, Clontech, Mountain View, CA), living colors mouse anti-GFP (632681, Clontech, Mountain View, CA), mouse anti-Rab11 (610656, BD Biosciences, Heidelberg, Germany), rabbit anti-Rab11a (ab65200, Abcam), rabbit anti-Rab11 (3539, Cell Signaling Technology, Danvers, MA), rabbit EGFR (4267, Cell Signaling Technology, Danvers, MA), rabbit pY1045 (2237, Cell Signaling Technology, Danvers, MA), rabbit pY1068 (3777, Cell Signaling Technology, Danvers, MA), goat EGFR (AF231, R&D Systems, Minneapolis, MN), mouse pY845 (558381, BD Biosciences, Heidelberg, Germany), mouse anti-Rab5 (610281, BD Biosciences, Heidelberg, Germany), rabbit anti-phospho ERK-1/2 Thr/Tyr 202/204 (9101, Cell Signaling Technology, Danvers, MA), mouse anti-ERK1/2 (Ab366991, Abcam); rabbit anti phosphor-Akt Ser473 (9271, Cell Signaling Technology, Danvers, MA), mouse anti-Akt (pan) (2920, Cell Signaling Technology, Danvers, MA).

### Secondary antibodies for immunofluorescence

Alexa Fluor® 647 chicken anti-rabbit IgG, Alexa Fluor® 647 donkey anti-mouse IgG, Alexa Fluor® 555 donkey anti-goat IgG, Alexa Fluor® 488 donkey anti-mouse IgG, Alexa Fluor® 546 goat anti-rabbit IgG, Alexa Fluor® 546 goat anti-mouse IgG (Life Technologies, Darmstadt, Germany).

### Secondary antibodies for Western blots

IRDye 680 donkey anti-mouse IgG, 800 donkey anti-rabbit IgG, (LI-COR Biosciences, Lincoln, NE).

## Molecular biology

mCitrine-N1, mCherry-N1, TagBFP-C1, and paGFP-N1 were generated by insertion of AgeI/BsrGI PCR fragments of mCitrine, mCherry, TagBFP (Evrogen, Moscow, Russia) and paGFP (gift from J. Lippincott-Schwartz) cDNA into pEGFP-C1 or -N1 (Clontech, Mountain View, CA). Fluorophore fusions of EGFR, PTB domain PTP1B, PTP1B D181A (gift from B. Neel, UHN Toronto) and Rab11a (Addgene) and Rab5 (gift from Y. Wu, CGC Dortmund) were generated through restriction-ligation of EGFR, PTB domain PTP1B, PTP1B D181A, Rab5, and Rab11a cDNA into the appropriate vector (mCherry-C1, Tagbfp-C1 and mCitrine-C1). Mutants of EGFR were generated by site-directed mutagenesis using the Quickchange Site-Directed-Mutagenesis kit (Stratagene, Santa Clara, CA). To generate EGFR-QG-mCitrine mCitrine flanked with a linker sequence (LAAAYSSILSSNLSSDS-mCitrine-

SDSSLNSSLISSYAAAL) (*Sabet et al., 2015*) was inserted between the amino acids Q958 and G959 of EGFR. LAMP1-TagBFP was generated by excision replacement of mTFP by TagBFP from LAMP1-mTFP (Allele Biotechnology and Pharmaceuticals, San Diego, CA) using the restriction sites MfeI and BamHI. Expression construct encoding fluorophore fusions of HA-c-Cbl were obtained by excision replacement of Citrine by mCherry from HA-c-Cbl-Citrine (gift from I.Dikic, iBC II, Frankfurt am Main). HA-Ubiquitin was a gift from I.Dikic. All constructs were sequence verified and tested for correct expression. pCMV-EGFR(128*)-EGFP-TCORS and p4-CMV-U6-PylT were described earlier (*Lang et al., 2012*).

## Reagents

Human EGF (Peprotech, Hamburg, Germany) was shock frozen at a concentration of 100 μg/ml in PBS + 0.1% BSA and stored at -−80°C. Site-specific *C*-terminal labeling of hEGF-Cys was carried out as described previously (*Sonntag et al., 2014*). In short, the unprotected *C*-terminal cysteine in hEGF-Cys was labeled with a five times excess of Alexa647-maleimide (Life Technologies, Darmstadt, Germany) in aqueous solution (PBS, pH 7.4) under argon for 2 hr at 4°C in the dark. Pervanadate was freshly prepared by adding sodium orthovanadate (S6508, Sigma Aldrich, St. Louis, MO) to $H_2O_2$ (30%) according to *Huyer et al. (1997)*). tet-TAMRA and TCOK were described earlier in *Lang et al. (2012)*.

## Cell culture, transfections, and site specific labeling of the extracellular EGFR domain

COS-7 and HEK293 cells were grown in Dulbecco's Modified Eagle's Medium (DMEM) supplemented with 10% fetal bovine serum (FBS), 2 mM L-glutamine and 1% non-essential amino acids (NEAA) and maintained at 37°C in 5% $CO_2$. Transfection of COS-7 cells was done using FUGENE6 (Roche Diagnostics, Mannheim, Germany) for imaging experiments and for Western blots according to manufacturer's protocol. Four to twelve hours prior to an experiment, cells were starved in growth medium containing dialyzed 0.5% FBS. For live cell microscopy, cells were cultured on 35-mm glass bottom dishes (MatTek, Ashland, MA) or 4-well chambered glass slides (Lab-tek, Thermo Fisher Scientific, Waltham, MA). HEK293 cells were plated on 25-mm poly-L-lysine coated coverslips in DMEM with 10% FBS. Transfection was done using lipofectamine 2000 reagent (Life Technologies, Darmstadt, Germany). The cells were incubated at 37°C with 5% $CO_2$ for 5 hr. After this period the media was replaced with 10% FBS DMEM containing 1 mM TCOK. After overnight incubation at 37°C in 5% $CO_2$, the cells were rinsed thrice in 0.5% FBS DMEM and incubated for at least 1 hr in 0.5% FBS DMEM. Afterwards, coverslips were inserted into a PFC pro-flow chamber (Warner instruments, Hamden, CT). Medium was flowed in by gravity flow. Labeling was carried out by flow through of 2 ml DMEM containing 400 nM tetrazine-TAMRA conjugate. This was followed by washout in DMEM or in DMEM containing EGF (80 ng/ml) or EGF-Alexa647 (20 ng/ml), respectively.

## RNA interference

Transfection of Rab11a siRNA was achieved using siRNA transfection reagent (sc-29528, Santa Cruz Biotechnology, Dallas, TX) in special transfection medium (sc-36868, Santa Cruz Biotechnology). Both Rab11a siRNA: sc-36340-SH (Santa Cruz Biotechnology) or scrambled non-targeting control siRNA: sc-37007 (Santa Cruz Biotechnology) was used at a final concentration of 50 nM for 72 hr before validation of knockdown.

Transfection of Rab11a/b siRNA was achieved using Lipofectamine 3000 Transfection Reagent (Thermo Fisher Scientific, Waltham, MA). Rab11a and Rab11b isoforms were simultaneously knocked down using double-stranded small interfering RNAs (siRNAs) targeted against the following target sequences 5′(AATGTCAGACAGACGCGAAAA)-3′ for Rab11a and 5′(AAGCACCTGACCTATGA-GAAC)-3′ for Rab11b at a final concentration of 40 nM for 72 hr before validation of knockdown. Scrambled non-targeting control siRNA (1027281, Qiagen) was used at a final concentration of 40 nM.

## Immunoprecipitation (IP) and western blotting (WB)

Cells were lysed in ready-made Cell Lysis Buffer (9803, Cell Signaling Technology, Danvers, MA) supplemented with Complete Mini EDTA-free protease inhibitor (Roche Applied Science, Heidelberg,

Germany) and 100 µl phosphatase inhibitor cocktail 2 and 3 (P5726 and P0044, Sigma Aldrich, St. Louis, MO). Following lysis, samples were cleared by centrifugation for 10 min, 13,000 rpm at 4°C. For ubiquitination IP, cell lysis was performed in modified RIPA (50 mM Tris–HCl, 150 mM NaCl, 1 mM EGTA, 1 mM EDTA, 1% Triton X-100, 1% Sodium deoxycholate, 0.2% SDS). Following lysis, samples were subjected to sonication for 12 s and then cleared by centrifugation for 15 min, 13,000 rpm at 4°C. Equal amounts of protein lysates were incubated with an anti-EGFR antibody over night at 4°C followed by incubation for 2 hr with Protein-G Sepharose® beads. SDS–PAGE was performed using the X-cell II mini electrophoresis apparatus (Life Technologies, Darmstadt, Germany) according to the manufacturers instructions. Samples were transferred to preactivated PVDF membranes (Merck Millipore, Billerica, MA) and incubated with the respective primary antibodies at 4°C overnight. Detection was performed using species-specific secondary IR-Dye 800 CW and IR-Dye 680 secondary antibodies (LI-COR Biosciences, Lincoln, NE) and the Odyssey Infrared Imaging System (LI-COR Biosciences, Lincoln, NE).

## Immunofluorescence

Cells were fixed with 4% paraformaldehyde in PBS (pH 7.4) for 10 min at room temperature (RT). This was followed by permeabilization for 5 min with 0.1% Triton X-100 in TBS. Background staining was blocked by incubation with Odyssey® Blocking Buffer (LI-COR Biosciences, Lincoln, NE) for 1 hr at RT. Primary and secondary antibodies (diluted in Odyssey® Blocking Buffer (LI-COR Biosciences, Lincoln, NE) were applied for 1 hr 30 min and 1 hr, respectively. Fixed cells were imaged in PBS at 37°C. In background-subtracted fluorescence images, masks for single cells were generated and the mean fluorescence intensity for each channel was measured in ImageJ (http://imagej.nih.gov/ij/). The relative phosphorylation level ($pY_n$/EGFR, n=845 $\bigcup$ 1045 $\bigcup$ 1068) per cell was determined and the mean values of $pY_n$/EGFR-mCitrine were calculated.

## Quantifying auto-phosphorylation from immunofluorescence and WB

The fluorescence signals emanating from the antibody complexes that were used to monitor phosphorylation of the three tyrosine residues cannot be compared because of the different affinities of the primary antibodies for the epitopes, and distinct fluorescence brightness and binding affinity of the labeled secondary antibodies. In order to be able to compare the level of ligand-independent phosphorylation for the three sites, the measured fraction of auto-phosphorylated EGFR (mean $pY_n$ intensity divided by mean EGFR intensity, where n=845,1045,1068) was normalized to the corresponding fraction after EGF stimulation ($pY_n$/EGFR-EGF upon 2' or 5' EGF stimulation: the maximally attainable phosphorylation at a given EGFR expression level). The relative EGFR auto-phosphorylation was computed from immunofluorescence for single cells and western blots as the ratio of the fraction of anti-$pY_n$ (n=845,1045,1068) intensity over EGFR-mCitrine intensity (for blots this corresponded to the anti-Citrine antibody band intensity) before and after stimulation with EGF. This 'ratio of fractions' gives a comparative measure for immunofluorescence and Western blots of the auto-phosphorylated fraction with respect to the EGF-induced phosphorylation of each investigated tyrosine residue on EGFR. For Western blots, the relative EGFR expression level was determined by normalizing to the average of the un-/stimulated EGFR-mCitrine bands of the highest levels of expressed EGFR-mCitrine (3 µg cDNA).

Quantification of the spatial distribution of spontaneously phosphorylated Y845 was performed by measuring the mean fluorescence intensity of EGFR-mCitrine and pY845 in a 5-pixel ring at the cell periphery. Mean fluorescence intensities of EGFR-mCitrine and pY845 at the RE were obtained by using binary masks of the RE, generated from thresholded BFP-Rab11a images. The difference between pY845 over EGFR-mCitrine at the PM and RE (($pY845/EGFR)_{PM}$ —$(pY845/EGFR)_{RE}$) was plotted as a function of overall EGFR phosphorylation level in individual cells.

## Confocal microscopy

Confocal images were recorded using an Olympus Fluoview FV1000 confocal microscope (Olympus Life Science Europa, Hamburg, Germany), a Leica TCS SP5 DMI6000 confocal microscope (Leica Microsystems, Wetzlar, Germany), a Leica SP8 confocal microscope (Leica Microsystems, Wetzlar, Germany) and a Zeiss LSM 780 (Carl Zeiss Jena GmbH, Jena, Germany).

## Olympus Fluoview FV100

The Olympus Fluoview™ FV1000 confocal microscope was equipped with a temperature controlled $CO_2$ incubation chamber at 37°C (EMBL, Heidelberg, Germany) and a 60x/1.35 NA Oil UPLSApo objective (Olympus, Hamburg, Germany). Fluorescent fusion proteins with BFP, mCitrine and mCherry or Alexa647-, Alexa555-, Alexa488-labeled secondary antibodies were excited using the 405 nm Diode-UV laser (FV5-LD05, Hatagaya), 488 nm Argon-laser (GLG 3135, Showa Optronics, Tokyo, Japan), 561 nm DPSS laser (85-YCA-020-230, Melles Griot, Bensheim, Germany) and 633 nm HeNe laser (05LHP-991, Melles Griot, Bensheim, Germany), respectively. Detection of fluorescence emission was restricted with an Acousto-Optical Beam Splitter (AOBS) as follows: BFP (425-–478 nm), mCitrine/Alexa488 (498-–551 nm), mCherry/Alexa555 (575-–675 nm), Alexa647 (655-–755 nm). In all cases, scanning was performed in frame-by-frame sequential mode with 2x frame averaging. The pinhole was set between 1.5 and 2.5 airy units.

## LeicaSP5 DMI6000

The Leica TCS SP5 DMI6000 confocal microscope was equipped with an environment-controlled chamber maintained at 37°C and a HCX PL APO 63x/1.4 NA oil objective (Leica Microsystems, Wetzlar, Germany). Fluorescent fusion proteins or Alexa-labeled secondary antibodies were excited using, a 405 nm Cube laser (1162002/AF, Coherent, Santa Clara, CA), 488 nm and 514 nm Argon laser line (LGK 7872 ML05, Lasos, Jena, Germany), a 561 nm DPSS laser (YLK 6120 T02, Lasos, Jena, Germany), and a 633 nm Helium-Neon laser line (LGK 7654–15, Lasos, Jena, Germany). Detection of fluorescence emission was restricted with an Acousto-Optical Beam Splitter (AOBS) as follows - BFP (415–458 nm), mCitrine (524–551 nm), mCherry (571–623 nm), and Alexa647 (643–680 nm). The pinhole was set to 250 µm or 350 µm and 16-bit images of 512x512 pixels were acquired.

## Leica SP8

The Leica TCS SP8 confocal microscope was equipped with an environment-controlled chamber (LIFE IMAGING SERVICES, Switzerland) maintained at 37°C and a HC PL APO CS2 1.4 NA oil objective (Leica Microsystems, Wetzlar, Germany). The fluorescent fusion proteins mCitrine and mCherry were excited using a 470–670 nm white light laser (white light laser Kit WLL2, NKT Photonics, Denmark) at 488 and 561 nm. Detection of fluorescence emission was restricted with an Acousto-Optical Beam Splitter (AOBS) as follows - —mCitrine (498-–551 nm) and mCherry at (575-–675 nm). The pinhole was set to 250 µm and 12-bit images of 512x512 pixels were acquired in sequential mode.

## TIRF Microscopy

Total Internal Reflection Fluorescence Microscopy (TIRFM) was performed on an Olympus IX81 microscope equipped with 60x NA = 1.8 TIRFM APOCHROMAT oil objective equipped with an EMCCD camera (C9100-13, Hamamatsu, Hersching, Germany). The microscope was coupled to an Argon and two diode lasers passing through a condenser that allows manipulation of the incident angle of the light onto the specimen. Wavelength selection was achieved through a Triple-pass 69000-ET-DAPI/FITC/TRITC filter set (Chroma), which is suitable for excitation with 405 nm, 488 nm, and 561 nm laser lines, and fluorescence imaging for TagBFP, mTFP, PAGFP, mCitrine, and mCherry fluorophores. After photo-activation (amplification of fluorescence intensity of EGFR-paGFP after irradiation with 405 nm for 500 ms) under TIRF-conditions at the basal PM of single COS7 cells co-transfected with Rab5-mCherry and EGFR-paGFP, a series of images was acquired for mCherry fluorescence and GFP fluorescence under TIRF-conditions, as well as by wide-field illumination with a focus adjusted 2 µm above the glass-substrate.

## Zeiss780

The Zeiss780 was equipped with a Plan apochromat 63x oil immersion objective. Imaging was performed with a scan resolution of 512x512 with 16x frame averaging. Fluorescent proteins were excited with diode 405 nm, Argon multiline, DPSS 561 nm and HeNe 633 nm laser lines. BFP was excited at 405 nm and detected at 415–530 nm, EGFP was excited at 488 nm and detected at 490–551 nm, TMR was excited at 561 nm and detected at 569–639 nm, and Alexa Fluor 647 was excited at 633 nm and detected at 638–755 nm, respectively.

## Quantification of the EGFR fraction at the RE and PM

Time-lapse movies of COS-7 cells expressing BFP-Rab11a, c-Cbl-mCherry and either EGFR-mCitrine wild type or ubiquitination-impaired mutants were obtained through confocal microscopy. Binary masks of the RE were generated from thresholded BFP-Rab11a images and for endosomal EGFR from thresholded EGFR-mCitrine images. To quantify the fraction of endosomal EGFR at the RE, the integrated EGFR-mCitrine fluorescence intensity was determined in the corresponding endosomal masks. The fraction of endosomal EGFR at the RE was determined by dividing the integrated intensity of EGFR at the RE by the integrated intensity of all endosomal EGFR. The initial ratio was normalized to 1 for each cell.

The fraction of EGFR at the PM (PM-EGFR) was quantified for individual cells as the ratio between the integrated intensity of a 5-pixel ring of the cell periphery and the integrated intensity of the whole cell. The ratio for time point 0min for each cell was normalized to 1 and mean values were calculated for each time point.

## FRAP, FLAP, and data analysis

FRAP experiments were carried out at 37°C on a Leica TCS SP5 DMI6000 confocal microscope equipped with a HCX PL APO 63x/1.4 NA oil objective and an environment-controlled chamber maintained at 37°C. The FRAP image acquisition was divided into three steps: (1) pre-bleach, (2) bleaching, and (3) post-bleach. In the pre-bleach step, a total of 10 fluorescence images for EGFR-mCitrine and BFP-Rab11a with a time interval of 10 s were acquired, using the 514 nm Argon laser at 10% power and the 405 nm Cube laser at 2% power, respectively. Bleaching of EGFR-mCitrine was performed with the 514 nm Argon laser at 100% power, and bleaching was restricted to a region of interest (ROI) on the RE, identified by expression of BFP-Rab11a. In the post-bleach step, a total of 80 fluorescence images for EGFR-mCitrine and BFP-Rab11a with a time interval of 10 s were acquired, using the 514 nm Argon laser at 10% power and the 405 nm Cube laser at 2% power, respectively. Images were background corrected in ImageJ and the relative fluorescence intensity *I(t)* was computed as follows:

$$I(t) = \left(\frac{I(t)_{ROI}}{I(t)_{TOTAL}}\right) / \left(\frac{I(0)_{ROI}}{I(0)_{TOTAL}}\right) \tag{1}$$

where $I(t)_{ROI}$ is the average fluorescence intensity in the ROI at time $t$, $I(t)_{TOTAL}$ is the average fluorescence intensity of the whole cell at the same time point, $I(0)_{ROI}$ is the average fluorescence intensity in the ROI of the pre-bleach images, $I(0)_{TOTAL}$ is the average fluorescence intensity of the whole cell of the pre-bleach images. The relative intensity was then fitted by a single exponential function:

$$I(t) = I_0 + A exp\left(\frac{-t}{\tau}\right) \tag{2}$$

where *I(t)* is the fluorescence intensity (in arbitrary units) at time *t*(s), $I_0$ is the residual intensity after photobleaching in the ROI, τ(s) is the exponential recovery time constant and *A* is the exponential amplitude.

FLAP experiments were carried out at 37°C on a Leica TCS SP5 DMI6000 confocal microscope equipped with a HCX PL APO 63x/1.4 NA oil objective. The FLAP image acquisition was divided into three steps: (1) pre-activation, (2) photoactivation and (3) post-activation. In the pre-activation step, a total of three fluorescence images for EGFR-paGFP and EGFR-mCherry with a time interval of 40 s were acquired, using the 488nm Argon laser at 10% power and the 561 nm DPSS laser at 11% power, respectively. Photoactivation of EGFR-paGFP was performed with the 405nm Cube laser at 80% power and photoactivation was restricted to a region of interest (ROI) on the RE, identified by co-expression of EGFR-mCherry. In the post-activation step, a total of 30 fluorescence images for EGFR-paGFP and EGFR-mCherry with a time interval of 40 s were acquired, using the 488 nm Argon laser at 10% power and the 561 nm DPSS laser at 11% power, respectively. At the end of the experiment an image of BFP-Rab11a was acquired with 5% 405-laser power. Following background correction, fluorescence loss after photoactivation at the RE was quantified as the ratio of local EGFR-paGFP to EGFR-mCherry fluorescence, to account for changes in the structure and intensity in the ROI. Fluorescence gain on the PM was quantified as the ratio of local (cell periphery) EGFR-paGFP to EGFR-mCherry fluorescence. Analogously to the FRAP experiment, exponential decay at the RE

and increase at the PM were fitted to a mono-exponential function (*Equation 2*). Considering the PM and the RE a simple two-compartment system allowed a more detailed interpretation of the retrieved parameters. The loss of EGFR from either compartment was assumed to follow a First-Order Rate Process. This yields the following exponential equation containing association ($k_{on}$) and dissociation ($k_{off}$) rate constants of the fluorescence intensity of EGFR to the RE($[EGFR_{RE}]$):

$$[EGFR_{RE}] = \left( \frac{k_{off}}{k_{on} + k_{off}} + \frac{k_{on}}{k_{on} + k_{off}} e^{-t(k_{on} + k_{off})} \right)$$ (3)

The normalized fluorescence decay curves were averaged and fitted to this function (*Equation 3*). The residence time of EGFR at the RE was calculated as 1/ $k_{off}$.

## Anisotropy microscopy

Anisotropy microscopy was done at 37°C in vitamin-free imaging medium in COS-7 cells ectopically expressing EGFR-QG-mCitrine and PTB-mCherry. Images were acquired 20–24 hr post-transfection, using an Olympus IX81 inverted microscope (Olympus, Hamburg, Germany) equipped with a MT20 illumination system. A linear dichroic polarizer (Meadowlark optics, Frederick, CO) was placed in the illumination path of the microscope, and two identical polarizers were placed in an external filter wheel at orientations parallel and perpendicular to the polarization of the excitation light. Fluorescence images were collected via a 20x/0.7 NA air objective using an Orca CCD camera (Hamamatsu Photonics, Hamamatsu City, Japan). For each measurement two images were taken, one with the emission polarizer oriented parallel to the excitation polarizer ($I_{\parallel}$) and one with the emission polarizer oriented perpendicular to the excitation polarizer ($I_{\perp}$). Steady state anisotropy ($r^i$) was calculated in each pixel *i* by:

$$r^i = \frac{G^i \, I_{\parallel}^i - I_{\perp}^i}{G^i \, I_{\parallel}^i - 2I_{\perp}^i}$$ (4)

The G-factor ($G^i$) was determined by acquiring the ratio of the intensities at perpendicular and parallel orientations for a fluorophore in solution (fluorescein) with a steady-state anisotropy close to zero. The CellR software supplied by the microscope manufacturer (Olympus, Hamburg, Germany) controlled data acquisition. Computed anisotropy images were displayed with false color-coding using ImageJ. For analysing the co-localization between EGFR-QG-mCitrine and PTB-mCherry the acquired fluorescence images were background subtracted and masks for individual cells were generated using ImageJ. PTB-mCherry fluorescence intensity was saturated in the nucleus and therefore excluded from the masks of individual cells. The extent of PTB recruitment (R) to EGFR was calculated by the following equation:

$$R = \frac{F \, of \, EGFR^{pre} * F \, of \, PTB^{post}}{F \, of \, EGFR^{post} * F \, of \, PTB^{pre}}$$ (5)

From this, a 2D-histogram of R versus EGFR fluorescence intensity per pixel was constructed with a bin size of 50 fluorescence units.

## FLIM and data analysis

FLIM measurements were made on an Olympus FluoView FV1000 laser scanning confocal microscope equipped with a time-correlated single-photon counting module (LSM Upgrade Kit, Picoquant, Berlin, Germany), using 60x/1.35 NA oil objective (Olympus, Hamburg, Germany). All pulsed lasers were controlled with the Sepia II software (PicoQuant GmbH, Berlin, Germany) at pulse repetition frequency of 40 MHz. The sample was excited using a 507 nm diode laser (LDH 507, Picoquant, Berlin, Germany). Fluorescence emission was spectrally filtered using a narrow-band emission filter (HQ 537/26, Chroma, Olching, Germany). Photons were detected using a single-photon counting avalanche photodiode (PDM Series, MPD, Picoquant, Berlin, Germany) and timed using a single-photon counting module (PicoHarp 300, Picoquant, Berlin, Germany). Using the SymPhoTime software V5.13 (Picoquant, Berlin, Germany), images were collected after an integration time of ~ 2 min collecting app. ~ 3.0–5.0 x $10^6$ photons. Data analysis was performed using the global analysis code as described in *Grecco et al. (2009)*.

## Computation of spatial coincidence

To quantify the spatial coincidence between the perinuclear area with high EGFR/PTP1B D181A interaction and the areas in the vicinity of Rab11-positive REs per cell, EGFR-mCtirine fluorescence lifetime and BFP-Rab11a fluorescence images were analyzed as follows:

1. Smoothening of EGFR-mCitrine fluorescence lifetime and BFP-Rab11a fluorescence intensity images to obtain an estimation of their densities by a consensus-like algorithm.
2. Using their respective density estimations, areas with significantly low EGFR-mCitrine lifetime and high BFP-Rab11a fluorescence intensity were identified using Otsu's method for multi-level thresholding (*Liao et al., 2001*). Tri-level thresholding was used to identify low EGFR-mCitrine lifetime areas and bi-level thresholding for identifying high BFP-Rab11a fluorescence intensity areas.
3. The overlap between the high BFP-Rab11a intensity areas and the perinuclear areas with low EGFR-mCitrine lifetime was represented as a percentage from the total area with low EGFR-mCitrine lifetime (*Figure 4E*).

The percentage of interacting EGFR/PTP1B D181A in the perinuclear area that coincided with areas of Rab11-rich endosomes was calculated from 20 cells (63 ± 29%, mean ± std.). This was compared to the expected overlap percentage, calculated as a ratio of the size of the high Rab11-intensity area to the total size of the cell (19 ± 12%).

Density estimation by a consensus-like algorithm: Given a snapshot image of a cell, from its pixel values we can observe how a quantity of interest (EGFR-mCitrine lifetime or BFP-Rab11a intensity) is spatially arranged. Due to fluctuations in the cell in each snapshot a non-smooth spatial arrangement is observed. Nevertheless, the images reveal some patterns, thus it can be presumed that each snapshot is a random realization of an underlying smooth density function. Estimating this function is of interest, as it can be easily compared to density functions of other quantities from the same cell.

For each pixel $i$, we denote with $N_i$ the set of its neighboring pixels from the cell and with $|N_i|$ the size of $N_i$, that is, the number of neighboring pixels of $i$. Typically $|N_i|$ is 4, except for the pixels that lie on the edges of the cell. Each pixel $i$ has an initial value $x_i(0)$, which depicts the value from the snapshot image. In order to make smoother density estimation, we aim to accommodate the differences between the values of neighboring pixels. Therefore, we assume that each pixel has a cost function of the form

$$C(x_i, x_{N_i}) = \frac{1}{2} \sum_{j \in N_i} \left( x_i - x_j \right)^2 + \frac{1}{2} K \left( x_i - x_j(0) \right)^2 \qquad (6)$$

that it tries to minimize. $K \geq 0$ is a parameter that represents the 'stubbornness' of every pixel regarding its initial value. Higher values of $K$ lead to larger stubbornness; hence, the final density estimate will be similar to the initial arrangement, thus non-smooth. For K = 0 consensus will be reached among the pixels, and all of them will have the same value in the final estimate. In our analysis, we use K = 0.005 to obtain a fairly smooth density estimate. An iterative consensus-like algorithm is applied for minimization of the cost function, similar to the one described in (*Ghaderi and Srikant, 2013*). For every pixel $i$ the estimate of the value for the next time step is

$$x_i(t+1) = \frac{1}{|N_i| + K} \sum_{j \in N_i} x_j(t) + \frac{K}{|N_i| + K} x_i(0) \qquad (7)$$

This algorithm converges exponentially fast to our desired density estimate.

## Statistical analysis

All results are expressed as mean ± SEM. If more than two groups were compared a one-way analysis of variance (ANOVA) was performed, followed by Tukeys multiple comparision test (Tukey's HSD) to estimate significance. For the comparison of the relative phosphorylation of Y845, Y1045, and Y1068 in *Figure 1B* via immunofluorescence a Z-test was performed. In case of comparisons with normalized data, an independent one sample $t$ test was performed.

## Mathematical modeling

The following set of ordinary differential equations:

r1 = k_endocyt EGFR^PM(t)

$r2 = k_{endocyt}\ EGFR^{PM}_p(t)$

$r3 = k_{recyc}\ EGFR^{int}(t)$

$r4 = k_{recyc}\ EGFR^{int}_p(t)$

$r5 = k^{PM}_{PTP}\ EGFR^{PM}_p(t)$

$r6 = k^{int}_{PTP}\ EGFR^{int}_p(t)$

$r7 = EGFR^{PM}(t)\ (k_{autonom}\ EGFR^{PM}(t) + k_{autocat}\ EGFR^{PM}_p(t))$

$r8 = EGFR^{int}(t)\ (k_{autonom}\ EGFR^{int}(t) + k_{autocat}\ EGFR^{int}_p(t))$

$k_{autocat} = 0\ OR\ k_{kinase}$

$k_{autonom} = 0.01\ k_{kinase}$

vesiscular trafficking / phosphorylation

$d/dt\ (EGFR^{PM}(t)) = - r1 + r3 + r5 - r7$

$d/dt\ (EGFR^{PM}_p(t)) = - r2 + r4 - r5 + r7$

$d/dt\ (EGFR^{int}(t)) = r1 - r3 + r6 - r8$

$d/dt\ (EGFR^{int}_p(t)) = r2 - r4 - r6 + r8$

$C = EGFR^{PM}(t) + EGFR^{PM}_p(t) + EGFR^{int}(t) + EGFR^{int}_p(t)$

were numerically integrated with the function 'NDSolve' of Mathematica 10.0.1 for Mac OS X. The following parameters yield the red curves in *Figure 6D*: $k_{endocyt}$=0.1/2, $k_{recyc}$=0.1/1, $k_{kinase}$=0.06, $k^{int}_{PTP}$ =0.0006*200, $k^{PM}_{PTP}$ =0.0006*15, C=2. Variations of these parameters leading to the other curves and images in *Figure 6* are indicated in the figure legends and in the main text, e.g. $k_{autocat}$=0 and $k_{PTP}$=0.0006 for the red curves in *Figure 6B*, or varying C in the x-axis of *Figure 6H*. For ligand binding (*Figure 6I*), the following kinase reaction rates were used:

$r7 = k_{ligand}\ EGF\text{-}EGFR^{PM}(t) / (EGF\text{-}EGFR^{PM}(t) + C_{saturation})$

$r8 = k_{ligand}\ EGF\text{-}EGFR^{int}(t) / (EGF\text{-}EGFR^{int}(t) + C_{saturation})$,

with $k_{ligand}$=0.06 and $C_{saturation}$=0.001. This results in essentially a zeroth order kinase reaction, without the numerical artifacts of a set of stiff ODEs for very small levels of phosphorylation $EGF\text{-}EGFR_p < C_{saturation}$.

## Acknowledgements

This project was partly supported by the following grants: HFSP grant no. RGP0039 (http://www.hfsp.org/funding/research-grants), the Max Planck Tandem project 'Molecular Activities in Liver Regeneration', and MRC Grants (U105181009 & UPA0241008). We thank Dr A Krämer and Dr A Koseska for critically reading this manuscript, A Stanoev for his help to analyse the EGFR-PTP1B D182A FLIM data and M Mahesh & K Lang for TCOK and tet-TAMRA.

## Additional information

### Funding

| Funder | Grant reference number | Author |
| --- | --- | --- |
| Human Frontier Science Program | RGP0039 | Ola Sabet |
| Max-Planck-Gesellschaft | | Georgia Xouri |
| Medical Research Council | U105181009 | Lloyd Davis |
| Medical Research Council | UPA0241008 | Jason W Chin |

The funders had no role in study design, data collection and interpretation, or the decision to submit the work for publication.

### Author contributions

MB, YB, Acquisition of data, Analysis and interpretation of data, Drafting or revising the article; MS, Developed the compartmental model and analysed FLAP and immunofluorescence data, Drafting or revising the article; GX, Acquisition of data, Analysis and interpretation of data; OS, Performed the FLAP and the Rab5 immunofluorescence experiments, Drafting or revising the article; LD, Designed and performed all experiments with non-natural amino acids; JWC, Supervised all experiments with

non-natural amino acids, Conception and design; PIHB, Conceived the project, Developed the compartmental model and analysed FLAP and immunofluorescence data, Drafting or revising the article

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
