## [Decision Letter]

Thank you for submitting your work entitled "EGF-dependent re-routing of vesicular recycling switches spontaneous phosphorylation suppression to EGFR signaling" for consideration by *eLife*. Your article has been reviewed by three peer reviewers, and the evaluation has been overseen by a Reviewing Editor and Tony Hunter as the Senior Editor. We hope you will be able to submit the revised version within a few days as the comments are very minor.

Summary:

This is a very high quality study that compares the consequences of auto-or EGF-induced phosphorylation of EGFR at 3 specific tyrosines in relation to EGFR trafficking, and combines this with recycling endosome and plasma membrane encounters with the tyrosine phosphatase PT1B to gain insight into how cells are able to maintain EGF responsiveness while retaining capacity to downregulate EGF-activated receptors. The authors have used great care to monitor these processes in relation to EGFR levels and the work will be of broad interest to scientists studying this medically important protein.

We recommend publication of the work in *eLife*.

Minor issues:

Figure 1A has only minimum and maximum values on X-axes, making it hard to understand.

The sentence "anti-tubulin was used as a loading control." starts with a lower-case letter a in Figure 1—figure Supplement 3, Figure 1—figure Supplement 4 and Figure 2—figure Supplement 1.

Is there a color symbol error in Figure 4D?

Results, subheading ‘Vesicular recycling suppresses spontaneous autocatalytic EGFR phosphorylation’: Rab11 depletion, please state if there was any change in total EGFR level in this condition. That Rab11 depletion influenced EGFR phosphorylation state is really important and should be highlighted in the discussion, as many workers fail to note such potentially pleiotropic consequences of Rab depletion.

[Editors’ note: a previous version of this study was rejected after peer review, but the authors submitted for reconsideration. The previous decision letter after peer review is shown below.]

Thank you for submitting your work entitled "EGF binding switches vesicular trafficking of EGFR from a suppressive cycle to a directional signaling mode" for peer review at *eLife*. Your submission has been carefully evaluated by Randy Schekman (Senior editor), a Reviewing editor, Marino Zerial and one other reviewer. Unfortunately, because of the numerous issues raised during the review process, we are not able to consider this version of the manuscript further. However, if you are able to address all of the issues raised by the reviewers, we would be willing to consider a resubmission of the story. In particular, it would be very important to explain more clearly, what is new for the field, at every stage of your story, and to include plasma membrane phosphatases in your presentation and conclusions.

I hope that you will find the comments useful in planning your next steps.

*Reviewer #1:*

There is a lot of work in this potentially interesting paper that highlights the following conclusions. 1. Autophosphorylation of EGFR favors Y845 rather than Y1045 and Y1068 docking sites; autophosphorylated EGFR appears to be monomeric by anisotropy; non-liganded EGFR cycles via Rab11 endosomes unless it engages Cbl and then is sent to lysosomes due to ubiquitylation; perinuclear protein tyrosine phosphatase dephosphorylated recycling endosome-localized EGFR. Expression of BFP-Rab11A may have decreased phospho-EGFR levels but it is not clear how significant these data are (is the difference driven by outliers?). To this non-EGFR expert, the data don't seem to bring forth a totally novel story except perhaps the different status of auto- and ligand-activated EGFRs. Without live cell microscopy, the authors can't make any conclusions about one EGFR class going directly to REs versus sorting in a common EE to ship cbl-engaged EGFRs to late endosomes and non-cbl engaged EGFRs to REs. The authors really need to highlight what is absolutely new in their analysis – the methods employed are sophisticated and provide interesting substance – even if they may not be totally ground breaking and may simply confirm previous conclusions obtained with less precise methods.

*Reviewer #2:*

The work by Baumdick et al. is devoted to study the differential trafficking behaviour of ligand-dependent vs. autonomously phosphorylated EGFR. Ligand-free phosphorylated EGFR is targeted to perinuclear recycling endosomes RE where high concentration of ER-bound phosphatase PTP1B ensures inactivation of the autonomously phosphorylated EGFR (spontaneous EGFR signalling). In contrast, the ligand-receptor complexes are targeted to degradation. It was previously demonstrated that targeting to degradation is triggered by ubiquitination of EGFR by c-Cbl. The authors demonstrate that c-Cbl is not recruited on activated Y1045 monomeric EGFR and, as such, activation on Y1045 is not sufficient to reroute EGFR to degradation. The spatial (non-homogeneous) distribution of PTP1B provides the explanation for the necessity of trafficking as a means to regulated EGFR phosphorylation. The study is very interesting. It shows that EGFR and ligand-occupied EGFR have very different trafficking fates, thus providing new insight into the regulation of EGFR trafficking in the cell that combines high sensitivity with robustness against "spurious" signaling.

As the manuscript is written, it gives the impression of a clear cut distribution of phosphatases between the plasma membrane and RE. We know that dephosphorylation occurs also at the PM (and the ER is located throughout the cell) and thus the authors need to take into account earlier findings and clarify the text throughout.

In the subsection “Ubiquitin-mediated switch in vesicular trafficking of ligand-activated EGFR”: The authors state "The translocation of EGFR to the PM after EGF addition coincided with a loss of EGFR fluorescence at the RE (Figure 4A, C). This shift in the steady-state distribution of EGFR occurs because recycling from the RE to the PM continues after EGF stimulation whereas vesicular trafficking to the RE is shut down." The authors imply that EGF stimulation transiently suppresses the "basal uptake" of EGFR. What is the evidence for this? The rerouting of this fraction of EGFR to degradation should lead to depletion of EGFR from RE, but I do not see how it should lead to a significant redistribution of EGFR toward the PM. Figure 4C, F provide quantitative evidence for depletion of EGFR from RE. However, the redistribution to PM is supported by one low-resolution image (see Figure 4A, 1 min vs. 0 min) of one cell without quantifications. The authors need to provide proper quantifications and statistics.

In the Discussion: "While EGF-bound receptor clusters await internalization, EGFR continues trafficking from the RE to the PM resulting in a perceived transient translocation of the receptor to the PM immediately after ligand stimulation (Figure 4A). This post-stimulus effect enhances the concentration of EGFR at the PM, further driving receptor dimerization and commitment to activation." Viera, Lamaze and Shmid, 1996, which the authors refer to showed that block of clathrin-dependent endocytosis does not change the basal uptake rate. This is opposite to what is claimed here. The authors should clarify or amend this statement. See comment above.

In the Discussion: "These EGF/EGFR complexes were rapidly internalized into Rab5 positive vesicles, whereas autonomously activated receptors did not co-localize with early endosomes (EE) (Figure 4—figure supplement 1). This suggests that autonomously activated EGFR directly traffics in clathrin coated vesicles from the PM to the RE (Vieira, Lamaze, and Schmid 1996)." This is an interesting and strong statement that needs to be well supported. I cannot judge the level of colocalization of EGFR to Rab5 from the images of Figure 4—figure supplement 1. The authors state that there is no colocalization of EGFR to Rab5 in the absence of EGF stimulation. However from Figure 4—figure supplement 1 one cannot draw this conclusion simply by qualitative observations. The magnification is too low, split colors presented as negatives, etc. The statement that internalized unoccupied EGFR is directly targeted to RE and bypass Rab5-positive EE has to be much better supported. I did not find the statement on direct traffic of clathrin coated vesicles to perinuclear RE in the referenced paper of Vieira, Lamaze, and Schmid 1996. Do the authors really mean that PM-derived vesicles (clathrin-dependent or clathrin-independent?) fuse directly with RE? This is not an established mechanism and the authors need to provide more convincing evidence or drop this statement.

Quantifications on Figure 4 and supplements are based on 3 to 10 cells (see legend). Given that error bar is SEM, what is the significance of reported differences? It has to be provided.

Figure 4—figure supplement 2 is not useful without quantifications. A and B are very similar to the naked eye.

*Reviewer #3:*

In the present manuscript the authors carry out the analysis of the regulation of EGFR signaling by endocytosis and receptor dephosphorylation using state-of-the-art methods of the quantitative fluorescence microscopy. The model proposed by the authors is in agreement with the current model of EGFR endocytic trafficking whereby in the absence of the ligand EGFR is constitutively internalized and recycled, whereas ligand activation of the receptor leads to ubiquitination and increased targeting of EGFR to the lysosomal degradation pathway. The authors further advance this model by proposing that constitutive endocytosis of ligand-free EGFR to Rab11 recycling compartment serves to counteract to the autocatalytic phosphorylation of EGFR in order to maintain low levels of the receptor activity. In essence, it is suggested that the tyrosine kinase of free, monomeric EGFR can be autonomously activated leading to receptor phosphorylation; however, endocytosis delivers free EGFR to the perinuclear area of the cell where an ER-associated phosphotyrosine phosphatase PTB-1B dephosphorylates the receptor prior to its recycling back to the cell surface. This new aspect of the EGFR endocytosis system is interesting but the supporting evidence is not sufficiently convincing.

1) The equilibrium between constitutive recycling and internalization results in a significantly larger pool of EGFR at the cell surface than in endosomes, at least when distribution of an endogenous EGFR is analyzed. If the assumptions in the present study are correct and endocytosis serves to maintain the endosomal pool of unoccupied EGFR in dephosphorylated state, the bulk of the receptor (located at the cell surface) would be capable of autonomous activation. Once recycled, there would be no mechanism in place to prevent re-activation of the receptors at the cell surface.

2) The key assumption supporting the proposed model that EGFR dephosphorylation takes place predominantly in the perinuclear area has been demonstrated in the previous study by this group. However, since the publication of the finding, it has been shown that transmembrane PTPs are essential for dephosphorylation of EGFR, and that these PTPs are located at the plasma membrane.

3) It has been shown in several publications that removal of EGF from the EGFR at the cell surface results in a rapid dephosphorylation of the receptor. If the authors believe that unoccupied EGFR is not dephosphorylated in the plasma membrane, this should be directly demonstrated.

4) Furthermore, the assumption that ER-associated PTP-1B exhibits a gradient of increasing activity from the plasma membrane to the perinuclear area is somewhat counterintuitive. ER spans the entire cell. The spatial proximities of the ER to the plasma membrane and recycling endosomes are similar. Likewise, ER forms numerous contacts with both the plasma membranes and endosomes.

It is rather disappointing that the key conclusions in the manuscript are reached in experiments performed under conditions of transient overexpression of EGFR, Rab11 and other constructs in cells that express endogenous EGFR or Rab11. It is established in the literature that the internalization rates of transiently overexpressed EGFR are low, and the regulatory mechanisms do not resemble what is observed in cells expressing endogenous or stably expressed EGFR, because of the saturation of the internalization and degradation pathways and other reasons. Similarly, overexpression of Rab11 dramatically changes the morphology of endosomal compartments and likely the trafficking itinerary of various cargo.

There are concerns with regards to the specific data and their interpretations, discussion of experiments and citing the literature as listed below.

1) In the first paragraph of the subsection “Autonomously and ligand-activated EGFR are different molecular states” with the reference to Figure 1B it is stated that Y1068 is less autonomously phosphorylated than Y845. This does not seem consistent with Figure 1B.

2) Figure 1E. It is unclear why after comparative analysis of pY1045 and pY1068, the experiments were switched to the use of the PTB domain that binds pY1148. The authors could easily use the SH2 domain of Grb2 that binds to pY1068. Further, PTB domain is accumulated in the perinuclear area suggesting that phosphorylated EGFR is accumulated in the same area. This data seems contradict the assumption that the receptor is predominantly dephosphorylated in the perinuclear area.

3) Figure 2A. It is impossible to resolve newly-synthesized EGFR from endocytosed EGFR the Golgi area by confocal microscopy. Overexpressed Rab11 is located in compartments with a strikingly morphology as compared to endogenous Rab11, which lessens confidence in the utility of the experiments with overexpressed Rab11.

4) Figure 2B. It is not apparent that photoactivated EGFR is delivered to the plasma membrane.

5) Figure 2C. These images clearly demonstrate that internalization of EGFR in the absence of the ligand is very slow (by 40 min, only a very small fraction is seen in punctate structures. These images also show that internalized free EGFR does not reach the recycling compartment marked by overexpressed Rab11. The signal corresponding to internalized EGFR surrounds this compartment.

6) Figure 3A. It is impossible to evaluate the extent of co-localization without the merge image and high resolution single-channel images.

7) Figure 3D. The image acquisition or image presentation parameters are set to emphasize intracellular compartments, and it is unclear whether these settings allow detection of the diffuse labeling of the surface EGFR. The co-localization of EGFR with Rab11 in these images is claimed to be 63%. This is not apparent from the visual comparison of quite different patterns of localization; here again, an overlap image is needed.

8) Figure 1—figure supplement 4. PTB-1B is highly overexpressed which makes it difficult to evaluate the extent of actual co-localization in this experiment.

9) Figure 2—figure supplement 1B. LAMP1 localization is highly unusual. This could be the effect of mistargeting due to the tag or overexpression.

10) Figure 3—figure supplement 1. These images are a clear example of an unusual localization likely due to accumulation of overexpressed proteins in the ER and/or other perturbed organelles.

11) Figure 4—figure supplements 1 and 4. These figures demonstrate that PV does not cause any significant endocytosis of ligand-free EGFR.

Discussion:

1) In the subsection “Ubiquitin-mediated switch in vesicular trafficking of ligand-activated EGFR”. The authors refer to one study that Cbl overexpression accelerates internalization. Multiple other studies clearly demonstrated that Cbl overexpression does not change the internalization rate but in increase the degradation rate of EGFR (in some cells).

2) The model in Figure 5 does not seem take into consideration recycling of EGF-EGFR complexes demonstrated by several groups.

3) The assumption that in the absence of the ligand, monomeric receptors are autonomously activated is questionable. There is still a debate in the literature about the existence of less stable unoccupied EGFR dimers. The FRET based assay used in the present study may not be sufficiently sensitive to formally disprove the existence of these dimers. In fact, the strong dependence of the receptor phosphorylation on its expression levels (Figure 1—figure supplement 1) suggests that it this autonomous phosphorylation correlates with the increased probability of monomer-monomer interaction.

---

## [Author Response]

Figure 1A has only minimum and maximum values on X-axes, making it hard to understand.

We have added the values on the X-axes of Figure 1A.

The sentence "anti-tubulin was used as a loading control." starts with a lower-case letter a in Figure 1—figure Supplement 3, Figure 1—figure Supplement 4 and Figure 2—figure Supplement 1.

This mistake was corrected.

Is there a color symbol error in Figure 4D?

Yes, indeed. We have amended the color symbol in Figure 4D.

*Results, subsection’ ‘Vesicular recycling suppresses spontaneous autocatalytic EGFR phosphorylation”: Rab11 depletion, please state if there was any change in total EGFR level in this condition. That Rab11 depletion influenced EGFR phosphorylation state is really important and should be highlighted in the discussion, as many workers fail to note such potentially pleiotropic consequences of Rab depletion.*

As suggested, we now also describe that Rab11a knock down had no effect on EGFR expression (Results, subsection’ ‘Vesicular recycling suppresses spontaneous autocatalytic EGFR phosphorylation”) and added the respective Western blot data in Figure 4—figure supplement 1. We have also added a sentence in the discussion on the pleiotropic effect of Rab knockdown on signaling activity in the Discussion.

[Editors’ note: the author responses to the previous round of peer review follow.]

Reviewer #1:

*There is a lot of work in this potentially interesting paper that highlights the following conclusions. 1. Autophosphorylation of EGFR favors Y845 rather than Y1045 and Y1068 docking sites; autophosphorylated EGFR appears to be monomeric by anisotropy; non-liganded EGFR cycles via Rab11 endosomes unless it engages Cbl and then is sent to lysosomes due to ubiquitylation; perinuclear protein tyrosine phosphatase dephosphorylated recycling endosome-localized EGFR. Expression of BFP-Rab11A may have decreased phospho-EGFR levels but it is not clear how significant these data are (is the difference driven by outliers?). To this non-EGFR expert, the data don't seem to bring forth a totally novel story except perhaps the different status of auto- and ligand-activated EGFRs. Without live cell microscopy, the authors can't make any conclusions about one EGFR class going directly to REs versus sorting in a common EE to ship cbl-engaged EGFRs to late endosomes and non-cbl engaged EGFRs to REs. The authors really need to highlight what is absolutely new in their analysis – the methods employed are sophisticated and provide interesting substance – even if they may not be totally ground breaking and may simply confirm previous conclusions obtained with less precise methods.*

We have now clearly described that the novelty of the paper is how the modulation of vesicular trafficking of EGFR by stimulus controls the signaling response of EGFR. We have also introduced the reason for investigating the phosphorylation of three tyrosine residues with the distinct functionalities of autocatalysis, signaling and trafficking. The data has now been presented in a more coherent way and new experiments further support our initial hypothesis that a switch in vesicular trafficking enables suppression of spontaneous activation while maintaining EGFRs signaling capacity. We have extended our life cell microscopy experiments to show that the different vesicular trafficking fates of recycling unliganded versus unidirectional trafficking of liganded EGFR are decided at the level of the EE (Figure 3). Importantly, we have now included a mathematical model (Figure 6) that explains how this switch in EGFR trafficking is a unique solution to suppress spontaneous autocatalysis while maintaining its signaling capacity.

Reviewer #2:*The work by Baumdick et al. is devoted to study the differential trafficking behaviour of ligand-dependent vs. autonomously phosphorylated EGFR. Ligand-free phosphorylated EGFR is targeted to perinuclear recycling endosomes RE where high concentration of ER-bound phosphatase PTP1B ensures inactivation of the autonomously phosphorylated EGFR (spontaneous EGFR signalling). In contrast, the ligand-receptor complexes are targeted to degradation. It was previously demonstrated that targeting to degradation is triggered by ubiquitination of EGFR by c-Cbl. The authors demonstrate that c-Cbl is not recruited on activated Y1045 monomeric EGFR and, as such, activation on Y1045 is not sufficient to reroute EGFR to degradation. The spatial (non-homogeneous) distribution of PTP1B provides the explanation for the necessity of trafficking as a means to regulated EGFR phosphorylation. The study is very interesting. It shows that EGFR and ligand-occupied EGFR have very different trafficking fates, thus providing new insight into the regulation of EGFR trafficking in the cell that combines high sensitivity with robustness against "spurious" signaling. As the manuscript is written, it gives the impression of a clear cut distribution of phosphatases between the plasma membrane and RE. We know that dephosphorylation occurs also at the PM (and the ER is located throughout the cell) and thus the authors need to take into account earlier findings and clarify the text throughout.*

We agree with the referee that EGFR dephosphorylation occurs at the PM, mainly by RPTPs. However, it has been demonstrated that RPTPs have a low catalytic efficiency and dephosphorylate signaling tyrosines. We have now incorporated our experimental findings into a mathematical model (Figure 6) that clearly defines the role of partitioning PTP activities in regulating EGFR autocatalytic phosphorylation. In short, low RPTP activity at the PM suffices to suppress autonomous phosphorylation of signaling tyrosines only because high PTP1B activity in the perinuclear area counters spontaneous autocatalysis by dephosphorylating tyrosine 845. We have now added a part in the Discussion to describe the different roles of plasma membrane localized RPTPs versus ER-associated PTPs that is fully consistent with our findings. In addition, we also discuss how suppression of spontaneous autocatalytic activation at the PM by regulated PTPs like Shp1/2 is incompatible with prolonged EGFR signaling.

*In the subsection “Ubiquitin-mediated switch in vesicular trafficking of ligand-activated EGFR”: The authors state "The translocation of EGFR to the PM after EGF addition coincided with a loss of EGFR fluorescence at the RE (Figure 4A, C). This shift in the steady-state distribution of EGFR occurs because recycling from the RE to the PM continues after EGF stimulation whereas vesicular trafficking to the RE is shut down." The authors imply that EGF stimulation transiently suppresses the "basal uptake" of EGFR. What is the evidence for this? The rerouting of this fraction of EGFR to degradation should lead to depletion of EGFR from RE, but I do not see how it should lead to a significant redistribution of EGFR toward the PM. Figure 4C, F provide quantitative evidence for depletion of EGFR from RE. However, the redistribution to PM is supported by one low-resolution image (see Figure 4A, 1 min vs. 0 min) of one cell without quantifications. The authors need to provide proper quantifications and statistics.*

Recycling of EGFR causes a steady-state population of EGFR to exist on the PM that is determined by the rates of anterograde and retrograde traffic. Upon EGF-stimulation we observe a net translocation to the PM that is explained by a delay in the anterograde internalization of EGFR while retrograde traffic from the RE remains unchanged. Ectopic c-Cbl expression enhances the rate of internalization, which reduces this net PM translocation, showing that ubiquitination might be a rate-limiting step in the ligand-mediated internalization of EGFR. We have now better described that this was concluded from the observed translocation of EGFR to the PM. We have also better visualized this translocation by mapping the differential of EGFR concentration as function of time (Figure 5B) and quantified the average PM translocation with and without ectopic c-Cbl expression (Figure 5C).

*In the Discussion: "While EGF-bound receptor clusters await internalization, EGFR continues trafficking from the RE to the PM resulting in a perceived transient translocation of the receptor to the PM immediately after ligand stimulation (Figure 4A). This post-stimulus effect enhances the concentration of EGFR at the PM, further driving receptor dimerization and commitment to activation." Viera, Lamaze and Shmid, 1996, which the authors refer to showed that block of clathrin-dependent endocytosis does not change the basal uptake rate. This is opposite to what is claimed here. The authors should clarify or amend this statement. See comment above.*

We agree with the referee and have removed this misleading reference. However, the translocation of EGFR after EGF-stimulus, which we have better quantified, can be explained by a delay in the endocytic uptake of EGFR. We have not further investigated the mode of endocytic uptake and removed our previous misleading statements. This has no implications on the presented mode of regulation of EGFR activity by vesicular trafficking.

*In the Discussion: "These EGF/EGFR complexes were rapidly internalized into Rab5 positive vesicles, whereas autonomously activated receptors did not co-localize with early endosomes (EE) (Figure 4—figure supplement 1). This suggests that autonomously activated EGFR directly traffics in clathrin coated vesicles from the PM to the RE (Vieira, Lamaze, and Schmid 1996)." This is an interesting and strong statement that needs to be well supported. I cannot judge the level of colocalization of EGFR to Rab5 from the images of Figure 4—figure supplement 1. The authors state that there is no colocalization of EGFR to Rab5 in the absence of EGF stimulation. However from Figure 4—figure supplement 1 one cannot draw this conclusion simply by qualitative observations. The magnification is too low, split colors presented as negatives, etc. The statement that internalized unoccupied EGFR is directly targeted to RE and bypass Rab5-positive EE has to be much better supported. I did not find the statement on direct traffic of clathrin coated vesicles to perinuclear RE in the referenced paper of Vieira, Lamaze, and Schmid 1996. Do the authors really mean that PM-derived vesicles (clathrin-dependent or clathrin-independent?) fuse directly with RE? This is not an established mechanism and the authors need to provide more convincing evidence or drop this statement.*

We thank the referee for pointing at this flaw in our interpretation of the experiments and now demonstrate that recycling as well as liganded receptors pass through Rab5 positive endosomes. This also means that ubiquitination reroutes liganded receptors after exiting the early endosomal compartment (Figures 3D, E, Figure 3—figure supplemental 2A, B).

*Quantifications on Figure 4 and supplements are based on 3 to 10 cells (see legend). Given that error bar is SEM, what is the significance of reported differences? It has to be provided.*

We have completely restructured most figures and included p-values as a metric for significance wherever it was appropriate. The content of our previous Figure 4 is now incorporated in Figure 5 and we make the point, that stimulation with EGF causes a significant translocation of EGFR (Figure 5A) with a net loss from the RE (Figure 5D). The quantifications are now more convincingly based on 6-14 cells, depending on the experiment.

*Figure 4—figure supplement 2 is not useful without quantifications. A and B are very similar to the naked eye.*

We repeated this experiment for n=9 cells for both wt and mutant EGFR and quantification of this important experiment is now Figure 5 H.

Reviewer #3:

*In the present manuscript the authors carry out the analysis of the regulation of EGFR signaling by endocytosis and receptor dephosphorylation using state-of-the-art methods of the quantitative fluorescence microscopy. The model proposed by the authors is in agreement with the current model of EGFR endocytic trafficking whereby in the absence of the ligand EGFR is constitutively internalized and recycled, whereas ligand activation of the receptor leads to ubiquitination and increased targeting of EGFR to the lysosomal degradation pathway. The authors further advance this model by proposing that constitutive endocytosis of ligand-free EGFR to Rab11 recycling compartment serves to counteract to the autocatalytic phosphorylation of EGFR in order to maintain low levels of the receptor activity. In essence, it is suggested that the tyrosine kinase of free, monomeric EGFR can be autonomously activated leading to receptor phosphorylation; however, endocytosis delivers free EGFR to the perinuclear area of the cell where an ER-associated phosphotyrosine phosphatase PTB-1B dephosphorylates the receptor prior to its recycling back to the cell surface. This new aspect of the EGFR endocytosis system is interesting but the supporting evidence is not sufficiently convincing.*

We thank the referee for his/her constructive comments and have addressed the relevant issues of overexpression. For this we have added two major experiments that convincingly support our model: Rab11 knock-down and quantification of ectopic in relation to endogenous EGFR levels. The former clearly shows that autocatalysis is suppressed by recycling and the latter that EGFR- mCitrine is expressed at about equal levels to endogenous EGFR.

*1) The equilibrium between constitutive recycling and internalization results in a significantly larger pool of EGFR at the cell surface than in endosomes, at least when distribution of an endogenous EGFR is analyzed. If the assumptions in the present study are correct and endocytosis serves to maintain the endosomal pool of unoccupied EGFR in dephosphorylated state, the bulk of the receptor (located at the cell surface) would be capable of autonomous activation. Once recycled, there would be no mechanism in place to prevent re-activation of the receptors at the cell surface.*

This non-intuitive concept of an effective PTP activity at the PM as a result of recycling has now been clarified by a theoretical model (Figure 6). Another non-intuitive aspect is that the spontaneous phosphorylation is a two-step process with slow kinetics determined by the thermal fluctuations in kinase conformation (“leaky kinase”) and rapid self-amplifying phosphorylation. If the amount of autonomously phosphorylated EGFR is high enough, autocatalytic phosphorylation takes over and greatly enhances phosphorylation kinetics. While the leaky kinase activity can be suppressed by PM-localized low PTP activity, the high PTP activity required to shut-down autocatalytic phosphorylation would also suppresses phosphorylation of EGF-stimulated EGFR at the PM. The solution of the system is to recycle EGFR to the perinuclear region with high PTP activity that also causes signal shut-down of internalized EGF-EGFR complexes. We now better describe that once the autocatalytic tyrosine is dephosphorylated, only slow kinase activity will rephosphorylate the receptor, which can be countered by low local RPTP activity. When the threshold for autocatalytic activation is passed, the system will rapidly phosphorylate and needs to be reset by dephosphorylation of Y845 by the high activity of perinuclear PTP1B.

*2) The key assumption supporting the proposed model that EGFR dephosphorylation takes place predominantly in the perinuclear area has been demonstrated in the previous study by this group. However, since the publication of the finding, it has been shown that transmembrane PTPs are essential for dephosphorylation of EGFR, and that these PTPs are located at the plasma membrane.*

We now stress in the manuscript the impact of EGFR-phosphorylation on different tyrosine (Figure 1), and that the autocatalytic site 845 is most sensitive to regulation by dephosphorylation at the perinuclear RE (Figure 4). Only by keeping phosphorylation of Y845 in check via recycling and preventing autocatalytic activation of EGFR, PM-associated PTPs like RPTPs can overcome the autonomous phosphorylation of signaling Y1068 and c-Cbl- binding Y1045. We have also added a discussion of the role of RPTPs versus ER-associated PTPs (see answer to the first comment of referee # 2).

*3) It has been shown in several publications that removal of EGF from the EGFR at the cell surface results in a rapid dephosphorylation of the receptor. If the authors believe that unoccupied EGFR is not dephosphorylated in the plasma membrane, this should be directly demonstrated.*

We have now clarified in the text that PTP activity at the PM can counter autonomous EGFR phosphorylation, but not autocatalytic phosphorylation. We demonstrate that in order to balance the autocatalytic activity of EGFR, the only solution is recycling through the perinuclear area. Indeed, inhibition of EGFR kinase activity by for example an ATP-binding site competitor results in dephosphorylation of signaling tyrosines (Offterdinger et al. 2004). In this case, there is no autocatalytic maintenance of the kinase activity and the PM-resident low PTP activity suffices to dephosphorylate EGFR on signaling tyrosines. We have better described that there is PTP activity at the PM that does not hinder an EGF response, while the high PTP activity necessary to suppress autocatalysis is in the perinuclear area in order to be compatible with EGFR signaling.

4) Furthermore, the assumption that ER-associated PTP-1B exhibits a gradient of increasing activity from the plasma membrane to the perinuclear area is somewhat counterintuitive. ER spans the entire cell. The spatial proximities of the ER to the plasma membrane and recycling endosomes are similar. Likewise, ER forms numerous contacts with both the plasma membranes and endosomes.

In previous work from this laboratory we have clearly demonstrated the gradient of increasing PTP1B activity towards the perinuclear area (Yudushkin et al., Science 2007). Moreover, based on the mass-action law, the strongest interaction between PTP1B and phosphorylated EGFR as its substrate should occur in the perinuclear area. This was substantiated here by showing that the interaction of EGFR with the trapping variant PTP1B-D181A mostly occurs in the perinuclear area within which the RE is contained (new Figure 5E). An important point is that EGFR on its way to the RE is exposed to the dephosphorylating PTP activity: only few of the PTP1B molecules happen to be in close proximity to the PM as compared to the bulk of PTP1B distributed over the ER. This low effective PTP1B activity at the PM can only dephosphorylate a small fraction of PM-localized EGFR.

*It is rather disappointing that the key conclusions in the manuscript are reached in experiments performed under conditions of transient overexpression of EGFR, Rab11 and other constructs in cells that express endogenous EGFR or Rab11. It is established in the literature that the internalization rates of transiently overexpressed EGFR are low, and the regulatory mechanisms do not resemble what is observed in cells expressing endogenous or stably expressed EGFR, because of the saturation of the internalization and degradation pathways and other reasons. Similarly, overexpression of Rab11 dramatically changes the morphology of endosomal compartments and likely the trafficking itinerary of various cargo.*

We agree with the referee and have now quantified the EGFR-mCitrine expression level relative to endogenous EGFR and show that it is of similar level. We have also performed two independent Rab11 knock-down experiments with the exciting and consistent result that it strongly augments the autocatalytic Y845 phosphorylation (~300%). The complementary perturbation by Rab11a-BFP ectopic expression enhances the partitioning of EGFR towards the RE and shows the opposite result of further suppressing Y845 phosphorylation. This clearly demonstrates that the function of recycling is suppression of spontaneous autocatalytic activation.

There are concerns with regards to the specific data and their interpretations, discussion of experiments and citing the literature as listed below.

*1) In the first paragraph of the subsection “Autonomously and ligand-activated EGFR are different molecular states” with the reference to Figure 1B it is stated that Y1068 is less autonomously phosphorylated than Y845. This does not seem consistent with Figure 1B.*

We have addressed this issue by quantifying western blot data as well as single cell immunofluorescence experiments (new Figure 1B) that both lead to the same conclusion: the c- Cbl binding site Y1045 is the least efficiently phosphorylated in the absence of EGF. However, pervanadate-mediated PTP-inhibition shows that spontaneous autacatalytic phosphorylation by the EGFR kinase activity indeed occurs most efficiently on Y845 as compared to Y1045 or Y1068 (new Figure 1C).

*2) Figure 1E. It is unclear why after comparative analysis of pY1045 and pY1068, the experiments were switched to the use of the PTB domain that binds pY1148. The authors could easily use the SH2 domain of Grb2 that binds to pY1068. Further, PTB domain is accumulated in the perinuclear area suggesting that phosphorylated EGFR is accumulated in the same area. This data seems contradict the assumption that the receptor is predominantly dephosphorylated in the perinuclear area.*

We use anisotropy to assess whether phosphorylated EGFR is monomeric or self-associated. In this experiment, binding of the PTB domain only serves as an indicator of EGFR phosphorylation on signaling tyrosines. To avoid confusion, we arranged this into the new Figure 2. The visible accumulation of PTB domain in the nuclear and perinuclear region is a result of widefield microscopy that integrates fluorescence in the z-direction of freely diffusible PTB domain. Figure 4—figure supplement 2 actually shows the interacting fraction of EGFR with the PTB domain and a substantial loss of interaction is clearly visible in the perinuclear area that includes the RE. The most important result of our anisotropy experiments is that upon PV treatment resulting in fully phosphorylated EGFR (new Figure 1C) no change in anisotropy could be observed in contrast to EGF stimulation (compare new Figure 2A with 2C).

*3) Figure 2A. It is impossible to resolve newly-synthesized EGFR from endocytosed EGFR the Golgi area by confocal microscopy. Overexpressed Rab11 is located in compartments with a strikingly morphology as compared to endogenous Rab11, which lessens confidence in the utility of the experiments with overexpressed Rab11.*

We performed colocalization experiments with Rab11a-BFP and EGFR-mCitrine after treatment with Cyclohexamide to block protein synthesis that showed no change in their colocalization (new Figure 3—figure supplement 1A).

*4) Figure 2B. It is not apparent that photoactivated EGFR is delivered to the plasma membrane.*

Ectopic Rab11a-BFP expression leads to a steady-state distribution of EGFR that is shifted toward the RE. This is clearly visible in the EGFR-mCherry fluorescence micrograph that shows a low basal PM staining (Figure 3B). EGFR steady state distribution dictates the low but significant signal of photoactivated EGFR that appears at the PM. We have now added a clearer example of this photoactivated EGFR-paGFP evolution towards the EGFR-mCherry steady state distribution.

*5) Figure 2C. These images clearly demonstrate that internalization of EGFR in the absence of the ligand is very slow (by 40 min, only a very small fraction is seen in punctate structures. These images also show that internalized free EGFR does not reach the recycling compartment marked by overexpressed Rab11. The signal corresponding to internalized EGFR surrounds this compartment.*

In order to incorporate a clickable unnatural amino acid into the extracellular part of EGFR at position 128, genetic code expansion was optimized in HEK293 cells. These cells exhibit slower recycling than COS-7 cells but the experiment clearly corroborates the anterograde traffic of EGFR from the PM to the RE. We included a fluorescence intensity profile across the RE in Figure 3—figure supplement 1 to show that the accumulated ring of TAMRA- fluorescence overlaps with the Rab11a-BFP signal.

*6) Figure 3A. It is impossible to evaluate the extent of co-localization without the merge image and high resolution single-channel images.*

For the sake of consistency and simplicity, we have replaced the FRET experiment with a ratiometric immunofluorescence experiment that clearly shows dephosphorylation of Y845 in the perinuclear area that encompasses the RE (new Figure 4C). The previous FRET experiment in life cells that corroborates this result has been removed.

*7) Figure 3D. The image acquisition or image presentation parameters are set to emphasize intracellular compartments, and it is unclear whether these settings allow detection of the diffuse labeling of the surface EGFR. The co-localization of EGFR with Rab11 in these images is claimed to be 63%. This is not apparent from the visual comparison of quite different patterns of localization; here again, an overlap image is needed.*

We actually computed the overlap between Rab11 and where EGFR interacts with PTP1B D/A. This is distinct from the overlap between EGFR and PTP1B. We have now included a quantification of average α in regions of high EGFR-mCitrine intensity as a function of mean fluorescence that shows that increasing the RE-biogenesis by ectopic Rab11a-BFP expression increases the interaction of EGFR with the PTP1B trapping mutant. This indicates a more efficient EGFR dephosphorylation upon shifting its steady state distribution towards the RE.

*8) Figure 1—figure supplement 4. PTB-1B is highly overexpressed which makes it difficult to evaluate the extent of actual co-localization in this experiment.*

This experiment shows the interaction of EGFR with PTB domain instead of PTP1B. Ratiometric quantification of PTB recruitment to EGFR was performed at the periphery of the cell, where there is no interference with the freely diffusible PTB domain in the nucleus. See also the answer to point 2).

*9) Figure 2—figure supplement 1B. LAMP1 localization is highly unusual. This could be the effect of mistargeting due to the tag or overexpression.*

We are not clear, what is unusual about a perinuclear staining of lysosomes by Lamp1-BFP in these small HEK293 cells. This experiment only shows that EGF-stimulated EGFR enters lysosomes within 2 hours.

*10) Figure 3—figure supplement 1. These images are a clear example of an unusual localization likely due to accumulation of overexpressed proteins in the ER and/or other perturbed organelles.*

This unusual localization is not due to accumulation of overexpressed proteins in the ER. Instead, the reversible trapping of EGFR on PTP1B-D181A causes an accumulation of both proteins in areas of their strongest interaction. This confirms that EGFR interacts most strongly with PTP1B on perinuclear membranes in the vicinity of the RE.

*11) Figure 4—figure supplements 1 and 4. These figures demonstrate that PV does not cause any significant endocytosis of ligand-free EGFR.*

Indeed we show that the steady state distribution caused by recycling does not substantially change by PV treatment, which is consistent with our model that spontaneously phosphorylated EGFR keeps recycling. However, we now also show that recycling receptors transit via Rab5 positive endosomes to the RE (new Figure 3D, E and Figure 3—figure supplement 2).

*Discussion: 1) In the subsection “Ubiquitin-mediated switch in vesicular trafficking of ligand-activated EGFR”. The authors refer to one study that Cbl overexpression accelerates internalization. Multiple other studies clearly demonstrated that Cbl overexpression does not change the internalization rate but in increase the degradation rate of EGFR (in some cells).*

We now show that in Cos-7 cells, Cbl expression clearly accelerates internalization (Figure 5C).

2) The model in Figure 5 does not seem take into consideration recycling of EGF-EGFR complexes demonstrated by several groups.

Recycling occurs mainly at low doses of EGF stimulation (Sigismund et al. 2005, PNAS 102(8), 2760–5), where some receptors might autocatalytically activate but fail to be ubiquitinated due to lack of stable EGF-induced self-association. In our study, where saturating doses of EGF were used, most receptors permanently internalized when we ectopically express c-Cbl. This shows that c-Cbl is a limiting factor in rerouting EGF-EGFR complexes to lysosomes. Therefore, the level of c-Cbl expression in different cells might dictate the fraction of EGF-EGFR complexes that keep recycling.

*3) The assumption that in the absence of the ligand, monomeric receptors are autonomously activated is questionable. There is still a debate in the literature about the existence of less stable unoccupied EGFR dimers. The FRET based assay used in the present study may not be sufficiently sensitive to formally disprove the existence of these dimers. In fact, the strong dependence of the receptor phosphorylation on its expression levels (Figure 1—figure supplement 1) suggests that it this autonomous phosphorylation correlates with the increased probability of monomer-monomer interaction.*

We embrace the idea of transient dimers that autonomously phosphorylate EGFR in trans based on second order rate reactions. This means that the fraction of interacting EGFR is very low at steady state. We are certain that our homo-FRET assay is sufficiently sensitive, because we measure a distinct drop in anisotropy upon EGF-stimulation (Figure 2A), in contrast to PV- mediated, autocatalytically fully phosphorylated EGFR (Figure 2C). This clearly shows that the autocatalytically phosphorylated EGFR is mostly monomeric.